# Solving Inverse Problems in Medical Imaging with Score-Based Generative Models

**Yang Song\*, Liyue Shen\*, Lei Xing & Stefano Ermon**
Stanford University
`{yangsong@cs,liyues@,lei@,ermon@cs}.stanford.edu`

## Abstract

Reconstructing medical images from partial measurements is an important inverse problem in Computed Tomography (CT) and Magnetic Resonance Imaging (MRI). Existing solutions based on machine learning typically train a model to directly map measurements to medical images, leveraging a training dataset of paired images and measurements. These measurements are typically synthesized from images using a fixed physical model of the measurement process, which hinders the generalization capability of models to unknown measurement processes. To address this issue, we propose a fully unsupervised technique for inverse problem solving, leveraging the recently introduced score-based generative models. Specifically, we first train a score-based generative model on medical images to capture their prior distribution. Given measurements and a physical model of the measurement process at test time, we introduce a sampling method to reconstruct an image consistent with both the prior and the observed measurements. Our method does not assume a fixed measurement process during training, and can thus be flexibly adapted to different measurement processes at test time. Empirically, we observe comparable or better performance to supervised learning techniques in several medical imaging tasks in CT and MRI, while demonstrating significantly better generalization to unknown measurement processes.

## 1 Introduction

Computed Tomography (CT) and Magnetic Resonance Imaging (MRI) are commonly used imaging tools for medical diagnosis. Reconstructing CT and MRI images from raw measurements (sinograms for CT and k-spaces for MRI) are well-known inverse problems. Specifically, measurements in CT are given by X-ray projections of an object from various directions, and measurements in MRI are obtained by inspecting the Fourier spectrum of an object with magnetic fields. However, since obtaining the full sinogram for CT causes excessive ionizing radiation for patients, and measuring the full k-space of MRI is very time-consuming, it has become important to reduce the number of measurements in CT and MRI. In many cases, only partial measurements, such as sparse-view sinograms and downsampled k-spaces, are available. Due to this loss of information, the inverse problems in CT and MRI are often ill-posed, making image reconstruction especially challenging.

With the rise of machine learning, many methods (Zhu et al., 2018; Mardani et al., 2017; Shen et al., 2019; Würfl et al., 2018; Ghani & Karl, 2018; Wei et al., 2020) have been proposed for medical image reconstruction using a small number of measurements. Most of these methods are supervised learning techniques. They learn to directly map partial measurements to medical images, by training on a large dataset comprising pairs of CT/MRI images and measurements. These measurements need to be synthesized from medical images with a fixed physical model of the measurement process. However, when the measurement process changes, such as using a different number of CT projections or different downsampling ratio of MRI k-spaces, we have to re-collect the paired dataset with the new measurement process and re-train the model. This prevents models from generalizing effectively to new measurement processes, leading to counter-intuitive instabilities such as more measurements causing worse performance (Antun et al., 2020).

---

\*Joint first authors.

In this work, we sidestep this difficulty completely by proposing unsupervised methods that do not require a paired dataset for training, and therefore are not restricted to a fixed measurement process. Our main idea is to learn the prior distribution of medical images with a generative model in order to infer the lost information due to partial measurements. Specifically, we propose to train a score-based generative model (Song & Ermon, 2019; 2020; Song et al., 2021) on medical images as the data prior, due to its strong performance in image generation (Ho et al., 2020; Dhariwal & Nichol, 2021). Given a trained score-based generative model, we provide a family of sampling algorithms to create image samples that are consistent with the observed measurements and the estimated data prior, leveraging the physical measurement process. Once our model is trained, it can be used to solve any inverse problem within the same image domain, as long as the mapping from images to measurements is linear, which holds for a large number of medical imaging applications.

We evaluate the performance of our method on several tasks in CT and MRI. Empirically, we observe comparable or better performance compared to supervised learning counterparts, even when evaluated with the same measurement process in their training. In addition, we are able to uniformly surpass all baselines when changing the number of measurements, *e.g.*, using a different number of projections in sparse-view CT or changing the k-space downsampling ratio in undersampled MRI. Moreover, we show that by plugging in a different measurement process, we can use a single model to perform both sparse-view CT reconstruction and metal artifact removal for CT imaging with metallic implants. To the best of our knowledge, this is the first time that generative models are reported successful on clinical CT data. Collectively, these empirical results indicate that our method is a competitive alternative to supervised techniques in medical image reconstruction and artifact removal, and has the potential to be a universal tool for solving many inverse problems within the same image domain.

## 2 BACKGROUND

### 2.1 LINEAR INVERSE PROBLEMS

An inverse problem seeks to recover an unknown signal from a set of observed measurements. Specifically, suppose $\mathbf{x} \in \mathbb{R}^n$ is an unknown signal, and $\mathbf{y} \in \mathbb{R}^m = \boldsymbol{A}\mathbf{x} + \boldsymbol{\epsilon}$ is a noisy observation given by $m$ linear measurements, where the measurement acquisition process is represented by a linear operator $\boldsymbol{A} \in \mathbb{R}^{m \times n}$, and $\boldsymbol{\epsilon} \in \mathbb{R}^n$ represents a noise vector. Solving a linear inverse problem amounts to recovering the signal $\mathbf{x}$ from its measurement $\mathbf{y}$. Without further assumptions, the problem is ill-defined when $m < n$, so we additionally assume that $\mathbf{x}$ is sampled from a prior distribution $p(\mathbf{x})$. In this probabilistic formulation, the measurement and signal are connected through a measurement distribution $p(\mathbf{y} \mid \mathbf{x}) = q_{\boldsymbol{\epsilon}}(\mathbf{y} - \boldsymbol{A}\mathbf{x})$, where $q_{\boldsymbol{\epsilon}}$ denotes the noise distribution of $\boldsymbol{\epsilon}$. Given $p(\mathbf{y} \mid \mathbf{x})$ and $p(\mathbf{x})$, we can solve the inverse problem by sampling from the posterior distribution $p(\mathbf{x} \mid \mathbf{y})$.

Examples of linear inverse problems in medical imaging include image reconstruction for CT and MRI. In both cases, the signal $\mathbf{x}$ is a medical image. The measurement $\mathbf{y}$ in CT is a sinogram formed by X-ray projections of the image from various angular directions (Buzug, 2011), while the measurement $\mathbf{y}$ in MRI consists of spatial frequencies in the Fourier space of the image (a.k.a. the k-space in the MRI community) (Vlaardingerbroek & Boer, 2013).

### 2.2 SCORE-BASED GENERATIVE MODELS

When solving inverse problems in medical imaging, we are given an observation $\mathbf{y}$, the measurement distribution $p(\mathbf{y} \mid \mathbf{x})$ and aim to sample from the posterior distribution $p(\mathbf{x} \mid \mathbf{y})$. The prior distribution $p(\mathbf{x})$ is typically unknown, but we can train generative models on a dataset $\{\mathbf{x}^{(1)}, \mathbf{x}^{(2)}, \cdots, \mathbf{x}^{(N)}\} \sim p(\mathbf{x})$ to estimate this prior distribution. Given an estimate of $p(\mathbf{x})$ and the measurement distribution $p(\mathbf{y} \mid \mathbf{x})$, the posterior distribution $p(\mathbf{x} \mid \mathbf{y})$ can be determined through Bayes' rule.

We propose to estimate the prior distribution of medical images using the recently introduced score-based generative models (Song & Ermon, 2019; Ho et al., 2020; Song et al., 2021), whose iterative sampling procedure makes it especially easy for controllable generation conditioned on an observation $\mathbf{y}$. Specifically, we adopt the formulation of score-based generative models in Song et al. (2021), where we leverage a Markovian diffusion process to progressively perturb data to noise, and then smoothly convert noise to samples of the data distribution by estimating and simulating its time reversal. We provide an illustration of this generative modeling framework in Fig. 1.

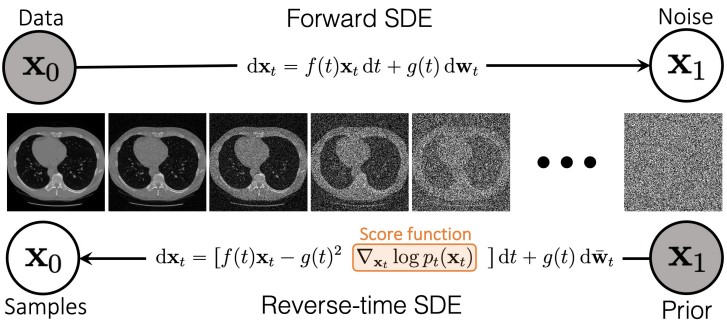

Figure 1: We can smoothly perturb images to noise by following the trajectory of an SDE. By estimating the score function $\nabla_{\mathbf{x}} \log p_t(\mathbf{x})$ with neural networks (called score models), it is possible to approximate the reverse SDE and then solve it to generate image samples from noise.

**Perturbation process** Suppose the dataset is sampled from an unknown data distribution $p(\mathbf{x})$. We perturb datapoints with a stochastic process over a time horizon $[0,1]$, governed by a linear stochastic differential equation (SDE) of the following form

$$\mathrm{d}\mathbf{x}_t = f(t)\mathbf{x}_t \, \mathrm{d}t + g(t) \, \mathrm{d}\mathbf{w}_t, \qquad t \in [0,1], \tag{1}$$

where $f : [0,1] \to \mathbb{R}$, $g : [0,1] \to \mathbb{R}$, $\{\mathbf{w}_t \in \mathbb{R}^n\}_{t \in [0,1]}$ denotes a standard Wiener process (a.k.a., Brownian motion), and $\{\mathbf{x}_t \in \mathbb{R}^n\}_{t \in [0,1]}$ symbolizes the trajectory of random variables in the stochastic process. We further denote the marginal probability distribution of $\mathbf{x}_t$ as $p_t(\mathbf{x})$, and the transition distribution from $\mathbf{x}_0$ to $\mathbf{x}_t$ as $p_{0t}(\mathbf{x}_t \mid \mathbf{x}_0)$. By definition, we clearly have $p_0(\mathbf{x}) \equiv p(\mathbf{x})$. Moreover, the functions $f(t)$ and $g(t)$ are specifically chosen such that for any initial distribution $p_0(\mathbf{x})$, the distribution at the end of the perturbation process, $p_1(\mathbf{x})$, is close to a pre-defined noise distribution $\pi(\mathbf{x})$. In addition, the transition density $p_{0t}(\mathbf{x}_t \mid \mathbf{x}_0)$ is always a conditional linear Gaussian distribution, taking the form $p_{0t}(\mathbf{x}_t \mid \mathbf{x}_0) = \mathcal{N}(\mathbf{x}_t \mid \alpha(t)\mathbf{x}_0, \beta^2(t)\boldsymbol{I})$ where $\alpha : [0,1] \to \mathbb{R}$ and $\beta : [0,1] \to \mathbb{R}$ can be derived analytically from $f(t)$ and $g(t)$ (Särkkä & Solin, 2019). Examples of such SDEs include Variance Exploding (VE), Variance Preserving (VP), and subVP SDEs proposed in Song et al. (2021). We found VE SDEs performed the best in our experiments.

**Reverse process** By reversing the perturbation process in Eq. (1), we can start from a noise sample $\mathbf{x}_1 \sim p_1(\mathbf{x})$ and gradually remove the noise therein to obtain a data sample $\mathbf{x}_0 \sim p_0(\mathbf{x}) \equiv p(\mathbf{x})$. Crucially, the time reversal of Eq. (1) is given by the following reverse-time SDE (Song et al., 2021)

$$\mathrm{d}\mathbf{x}_t = \left[ f(t)\mathbf{x}_t - g(t)^2 \nabla_{\mathbf{x}_t} \log p_t(\mathbf{x}_t) \right] \mathrm{d}t + g(t) \, \mathrm{d}\bar{\mathbf{w}}_t, \qquad t \in [0,1], \tag{2}$$

where $\{\bar{\mathbf{w}}_t\}_{t \in [0,1]}$ denotes a standard Wiener process in the reverse-time direction, and $\mathrm{d}t$ represents an infinitesimal negative time step, since the above SDE must be solved backwards from $t = 1$ to $t = 0$. The quantity $\nabla_{\mathbf{x}_t} \log p_t(\mathbf{x}_t)$ is known as the *score function* of $p_t(\mathbf{x}_t)$. By the definition of time reversal, the trajectory of the reverse stochastic process given by Eq. (2) is $\{\mathbf{x}_t\}_{t \in [0,1]}$, same as the one from the forward SDE in Eq. (1).

**Sampling** Given an initial sample from $p_1(\mathbf{x})$, as well as scores at each intermediate time step, $\nabla_{\mathbf{x}} \log p_t(\mathbf{x})$, we can simulate the reverse-time SDE in Eq. (2) to obtain samples from the data distribution $p_0(\mathbf{x}) \equiv p(\mathbf{x})$. In practice, the initial sample is approximately drawn from $\pi(\mathbf{x})$ since $\pi(\mathbf{x}) \approx p_1(\mathbf{x})$, and the scores are estimated by training a neural network $s_{\boldsymbol{\theta}}(\mathbf{x}, t)$ (named the *score model*) on a dataset $\{\mathbf{x}^{(1)}, \mathbf{x}^{(2)}, \cdots, \mathbf{x}^{(N)}\} \sim p(\mathbf{x})$ with denoising score matching (Vincent, 2011; Song et al., 2021), *i.e.*, solving the following objective

$$\boldsymbol{\theta}^* = \arg\min_{\boldsymbol{\theta}} \frac{1}{N} \sum_{i=1}^{N} \mathbb{E}_{t \sim \mathcal{U}[0,1]} \mathbb{E}_{\mathbf{x}_t^{(i)} \sim p_{0t}(\mathbf{x}_t^{(i)} \mid \mathbf{x}^{(i)})} \left[ \left\| s_{\boldsymbol{\theta}}(\mathbf{x}_t^{(i)}, t) - \nabla_{\mathbf{x}_t^{(i)}} \log p_{0t}(\mathbf{x}_t^{(i)} \mid \mathbf{x}^{(i)}) \right\|_2^2 \right],$$

where $\mathcal{U}[0,1]$ denotes a uniform distribution over $[0,1]$. The theory of denoising score matching ensures that $s_{\boldsymbol{\theta}*}(\mathbf{x}, t) \approx \nabla_{\mathbf{x}} \log p_t(\mathbf{x})$. After training this score model, we plug it into Eq. (2) and solve the resulting reverse-time SDE

$$\mathrm{d}\mathbf{x}_t = \left[ f(t)\mathbf{x}_t - g(t)^2 s_{\boldsymbol{\theta}*}(\mathbf{x}_t, t) \right] \mathrm{d}t + g(t) \, \mathrm{d}\bar{\mathbf{w}}_t, \qquad t \in [0,1], \tag{3}$$

for sample generation. One sampling method is to use the Euler-Maruyama discretization for solving Eq. (3), as given in Algorithm 1. Other sampling methods include annealed Langevin dynamics (ALD, Song & Ermon, 2019), probability flow ODE solvers (Song et al., 2021), and Predictor-Corrector samplers (Song et al., 2021).

| **Algorithm 1** Unconditional sampling | **Algorithm 2** Inverse problem solving |
|---|---|
| **Require:** $N$ | **Require:** $N, \mathbf{y}, \lambda$ |
| 1: $\hat{\mathbf{x}}_1 \sim \pi(\mathbf{x}), \Delta t \leftarrow \frac{1}{N}$ | 1: $\hat{\mathbf{x}}_1 \sim \pi(\mathbf{x}), \Delta t \leftarrow \frac{1}{N}$ |
| 2: **for** $i = N - 1$ **to** $0$ **do** | 2: **for** $i = N - 1$ **to** $0$ **do** |
| 3: $\quad t \leftarrow \frac{i+1}{N}$ | 3: $\quad t \leftarrow \frac{i+1}{N}$ |
| | 4: $\quad \hat{\mathbf{y}}_t \sim p_{0t}(\mathbf{y}_t \mid \mathbf{y})$ |
| | 5: $\quad \hat{\mathbf{x}}_t \leftarrow \boldsymbol{T}^{-1}[\lambda \boldsymbol{\Lambda} \mathcal{P}^{-1}(\boldsymbol{\Lambda})\hat{\mathbf{y}}_t + (1-\lambda)\boldsymbol{\Lambda}\boldsymbol{T}\hat{\mathbf{x}}_t + (\boldsymbol{I} - \boldsymbol{\Lambda})\boldsymbol{T}\hat{\mathbf{x}}_t]$ |
| 4: $\quad \hat{\mathbf{x}}_{t-\Delta t} \leftarrow \hat{\mathbf{x}}_t - f(t)\hat{\mathbf{x}}_t \Delta t$ | 6: $\quad \hat{\mathbf{x}}_{t-\Delta t} \leftarrow \hat{\mathbf{x}}_t - f(t)\hat{\mathbf{x}}_t \Delta t$ |
| 5: $\quad \hat{\mathbf{x}}_{t-\Delta t} \leftarrow \hat{\mathbf{x}}_{t-\Delta t} + g(t)^2 \boldsymbol{s}_{\boldsymbol{\theta}*}(\hat{\mathbf{x}}_t, t)\Delta t$ | 7: $\quad \hat{\mathbf{x}}_{t-\Delta t} \leftarrow \hat{\mathbf{x}}_{t-\Delta t} + g(t)^2 \boldsymbol{s}_{\boldsymbol{\theta}*}(\hat{\mathbf{x}}_t, t)\Delta t$ |
| 6: $\quad \mathbf{z} \sim \mathcal{N}(\mathbf{0}, \boldsymbol{I})$ | 8: $\quad \mathbf{z} \sim \mathcal{N}(\mathbf{0}, \boldsymbol{I})$ |
| 7: $\quad \hat{\mathbf{x}}_{t-\Delta t} \leftarrow \hat{\mathbf{x}}_{t-\Delta t} + g(t)\sqrt{\Delta t}\,\mathbf{z}$ | 9: $\quad \hat{\mathbf{x}}_{t-\Delta t} \leftarrow \hat{\mathbf{x}}_{t-\Delta t} + g(t)\sqrt{\Delta t}\,\mathbf{z}$ |
| 8: **return** $\hat{\mathbf{x}}_0$ | 10: **return** $\hat{\mathbf{x}}_0$ |

# 3 SOLVING INVERSE PROBLEMS WITH SCORE-BASED GENERATIVE MODELS

With score-based generative modeling, we can train a score model $\boldsymbol{s}_{\boldsymbol{\theta}*}(\mathbf{x}, t)$ to generate unconditional samples from the the prior distribution of medical images $p(\mathbf{x})$. To solve inverse problems however, we will need to sample from the posterior $p(\mathbf{x} \mid \mathbf{y})$. This can be accomplished by conditioning the original stochastic process $\{\mathbf{x}_t\}_{t \in [0,1]}$ on an observation $\mathbf{y}$, yielding a *conditional* stochastic process $\{\mathbf{x}_t \mid \mathbf{y}\}_{t \in [0,1]}$. We denote the marginal distribution at $t$ as $p_t(\mathbf{x}_t \mid \mathbf{y})$, and our goal is to sample from $p_0(\mathbf{x}_0 \mid \mathbf{y})$, the same distribution as $p(\mathbf{x} \mid \mathbf{y})$ by definition. Much like generating unconditional samples by solving the reverse-time SDE in Eq. (2), we can reverse the conditional stochastic process $\{\mathbf{x}_t \mid \mathbf{y}\}_{t \in [0,1]}$ to sample from the posterior distribution $p_0(\mathbf{x}_0 \mid \mathbf{y})$ by solving the following *conditional* reverse-time SDE (Song et al., 2021):

$$\mathrm{d}\mathbf{x}_t = \left[ f(t)\mathbf{x}_t - g(t)^2 \nabla_{\mathbf{x}_t} \log p_t(\mathbf{x}_t \mid \mathbf{y}) \right] \mathrm{d}t + g(t)\,\mathrm{d}\bar{\mathbf{w}}_t, \qquad t \in [0,1]. \tag{4}$$

The conditional score function $\nabla_{\mathbf{x}_t} \log p_t(\mathbf{x}_t \mid \mathbf{y})$ is a critical part of Eq. (4), yet it is non-trivial to compute. One solution is to estimate the score function by training a new score model $\boldsymbol{s}_{\boldsymbol{\theta}*}(\mathbf{x}_t, \mathbf{y}, t)$ that explicitly depends on $\mathbf{y}$ (Song et al., 2021; Dhariwal & Nichol, 2021), such that $\boldsymbol{s}_{\boldsymbol{\theta}*}(\mathbf{x}_t, \mathbf{y}, t) \approx \nabla_{\mathbf{x}_t} \log p_t(\mathbf{x}_t \mid \mathbf{y})$. However, this requires paired data $\{(\mathbf{x}_i, \mathbf{y}_i)\}_{i=1}^N$ for training and has the same drawbacks as supervised learning techniques. We do not consider this approach in this work.

An unsupervised alternative is to approximate the conditional score function with an unconditionally-trained score model $\boldsymbol{s}_{\boldsymbol{\theta}*}(\mathbf{x}_t, t) \approx \nabla_{\mathbf{x}_t} \log p_t(\mathbf{x}_t)$ and the measurement distribution $p(\mathbf{y} \mid \mathbf{x})$. Many existing works (Song et al., 2021; Kawar et al., 2021; Kadkhodaie & Simoncelli, 2020; Jalal et al., 2021) have implemented this idea in different ways. However, the methods in Kawar et al. (2021) and Kadkhodaie & Simoncelli (2020) both require computing the singular value decomposition (SVD) of $\boldsymbol{A} \in \mathbb{R}^{m \times n}$, which can be difficult for many measurement processes in medical imaging. The method proposed in Jalal et al. (2021) is only designed for a specific sampling method called annealed Langevin dynamics (ALD, Song & Ermon, 2019), which proves to be inferior to more advanced sampling algorithms such as Predictor-Corrector methods (Song et al., 2021).

In what follows, we propose a new conditional sampling approach for inverse problem solving with score-based generative models. Our method is computationally efficient for medical image reconstruction, and is applicable to a large family of iterative sampling methods for score-based generative models. At a high level, we first train an unconditional score model $\boldsymbol{s}_{\boldsymbol{\theta}*}(\mathbf{x}, t)$ on medical images without assuming any measurement process. Given an observation $\mathbf{y}$ at test time, we form a stochastic process $\{\mathbf{y}_t\}_{t \in [0,1]}$ by adding appropriate noise to $\mathbf{y}$. We then discretize the reverse-time SDE in Eq. (3) with existing unconditional samplers for $\boldsymbol{s}_{\boldsymbol{\theta}*}(\mathbf{x}, t)$, while incorporating the conditional information from $\mathbf{y}$ with a proximal optimization step to generate intermediate samples that are consistent with $\{\mathbf{y}_t\}_{t \in [0,1]}$.

## 3.1 A CONVENIENT FORM OF THE LINEAR MEASUREMENT PROCESS

Many different measurement processes in medical imaging share same components of computation. For example, sparse-view CT reconstruction and metal artifact removal for CT both involve computing the same Radon transform. Similarly, MRI measurement processes require computing the same spatial Fourier transform regardless of different downsampling ratios. To rigorously characterize this structure of measurement processes, we propose a special formulation of $\boldsymbol{A}$ that is efficient to obtain

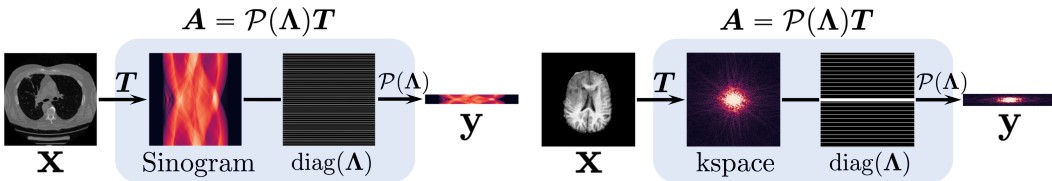

Figure 2: Linear measurement processes for sparse-view CT (*left*) and undersampled MRI (*right*).

in medical imaging applications. Without loss of generality, we assume that the linear operator $\boldsymbol{A}$ has full rank, *i.e.*, $\mathrm{rank}(\boldsymbol{A}) = \min(n, m) = m$. The result below gives the alternative formulation of $\boldsymbol{A}$:

**Proposition 1.** *If* $\mathrm{rank}(\boldsymbol{A}) = m$, *then there exist an invertible matrix* $\boldsymbol{T} \in \mathbb{R}^{n \times n}$, *and a diagonal matrix* $\boldsymbol{\Lambda} \in \{0, 1\}^{n \times n}$ *with* $\mathrm{tr}(\boldsymbol{\Lambda}) = m$, *such that* $\boldsymbol{A} = \mathcal{P}(\boldsymbol{\Lambda})\boldsymbol{T}$. *Here* $\mathcal{P}(\boldsymbol{\Lambda}) \in \{0, 1\}^{m \times n}$ *is an operator that, when multiplied with any vector* $\boldsymbol{a} \in \mathbb{R}^n$, *reduces its dimensionality to* $m$ *by removing each $i$-th element of* $\boldsymbol{a}$ *for* $i = 1, 2, \cdots, n$ *if* $\boldsymbol{\Lambda}_{ii} = 0$.

We illustrate this decomposition for CT/MRI in Fig. 2. Many measurement processes in medical imaging share the same $\boldsymbol{T}$, even if they correspond to different $\boldsymbol{A}$. For example, $\boldsymbol{T}$ corresponds to the Radon transform and Fourier transform in sparse-view CT and undersampled MRI respectively, regardless of the number of measurements, *i.e.*, CT projections and k-space downsampling ratios. For both sparse-view CT reconstruction and metal artifact removal for CT images, the operator $\boldsymbol{T}$ is the Radon transform (see Fig. 8). Intuitively, $\mathrm{diag}(\boldsymbol{\Lambda})$ can be viewed as a subsampling mask on the sinogram/k-space, and $\mathcal{P}(\boldsymbol{\Lambda})$ subsamples the sinogram/k-space into an observation $\mathbf{y}$ with a smaller size according to this subsampling mask. In addition, we note that $\boldsymbol{T}^{-1}$ can be efficiently implemented with the inverse Radon transform or the inverse Fourier transform in CT/MRI applications.

## 3.2 INCORPORATING A GIVEN OBSERVATION INTO AN UNCONDITIONAL SAMPLING PROCESS

In what follows, we show that the decomposition in Proposition 1 provides an efficient way to generate approximate samples from the conditional stochastic process $\{\mathbf{x}_t \mid \mathbf{y}\}_{t \in [0,1]}$ with an *unconditional* score model $\boldsymbol{s}_{\boldsymbol{\theta}*}(\mathbf{x}, t)$. The basic idea is to "hijack" the unconditional sampling process of score-based generative models to incorporate an observed measurement $\mathbf{y}$.

As we have already discussed, it is difficult to directly solve $\{\mathbf{x}_t \mid \mathbf{y}\}_{t \in [0,1]}$ for sample generation. To bypass this difficulty, we first consider a related stochastic process that is much easier to sample from. Recall that $p_{0t}(\mathbf{x}_t \mid \mathbf{x}_0) = \mathcal{N}(\mathbf{x}_t \mid \alpha(t)\mathbf{x}_0, \beta^2(t)\boldsymbol{I})$ where $\alpha(t)$ and $\beta(t)$ can be derived from $f(t)$ and $g(t)$ (Song et al., 2021). Given the unconditional stochastic process $\{\mathbf{x}_t\}_{t \in [0,1]}$, we define $\{\mathbf{y}_t\}_{t \in [0,1]}$, where $\mathbf{y}_t = \boldsymbol{A}\mathbf{x}_t + \alpha(t)\boldsymbol{\epsilon}$. Unlike $\{\mathbf{x}_t \mid \mathbf{y}\}_{t \in [0,1]}$, the conditional stochastic process $\{\mathbf{y}_t \mid \mathbf{y}\}_{t \in [0,1]}$ is fully tractable. First, we have $\mathbf{y}_0 = \boldsymbol{A}\mathbf{x}_0 + \alpha(0)\boldsymbol{\epsilon} = \boldsymbol{A}\mathbf{x}_0 + \boldsymbol{\epsilon} = \mathbf{y}$. Since $p_{0t}(\mathbf{x}_t \mid \mathbf{x}_0) = \mathcal{N}(\mathbf{x}_t \mid \alpha(t)\mathbf{x}_0, \beta^2(t)\boldsymbol{I})$, we have $\mathbf{x}_t = \alpha(t)\mathbf{x}_0 + \beta(t)\mathbf{z}$, where $\mathbf{z} \in \mathbb{R}^n \sim \mathcal{N}(\mathbf{0}, \boldsymbol{I})$. By definition, $\mathbf{y}_t = \boldsymbol{A}\mathbf{x}_t + \alpha(t)\boldsymbol{\epsilon}$, so we have $\mathbf{y}_t = \boldsymbol{A}(\alpha(t)\mathbf{x}_0 + \beta(t)\mathbf{z}) + \alpha(t)\boldsymbol{\epsilon} = \alpha(t)(\mathbf{y} - \boldsymbol{\epsilon}) + \beta(t)\boldsymbol{A}\mathbf{z} + \alpha(t)\boldsymbol{\epsilon} = \alpha(t)\mathbf{y} + \beta(t)\boldsymbol{A}\mathbf{z}$. Therefore, we can easily generate a sample $\hat{\mathbf{y}}_t \sim p_t(\mathbf{y}_t \mid \mathbf{y})$ by first drawing $\mathbf{z} \sim \mathcal{N}(\mathbf{0}, \boldsymbol{I})$ and then computing $\hat{\mathbf{y}}_t = \alpha(t)\mathbf{y} + \beta(t)\boldsymbol{A}\mathbf{z}$.

The key of our approach is to modify any existing iterative sampling algorithm designed for the unconditional stochastic process $\{\mathbf{x}_t\}_{t \in [0,1]}$ so that the samples are consistent with $\{\mathbf{y}_t \mid \mathbf{y}\}_{t \in [0,1]}$. In general, an iterative sampling process of score-based generative models selects a sequence of time steps $\{0 = t_0 < t_1 < \cdots < t_N = 1\}$ and iterates according to

$$\hat{\mathbf{x}}_{t_{i-1}} = \boldsymbol{h}(\hat{\mathbf{x}}_{t_i}, \mathbf{z}_i, \boldsymbol{s}_{\boldsymbol{\theta}*}(\hat{\mathbf{x}}_{t_i}, t_i)), \quad i = N, N-1, \cdots, 1, \tag{5}$$

where $\hat{\mathbf{x}}_{t_N} \sim \pi(\mathbf{x})$, $\mathbf{z}_i \sim \mathcal{N}(\mathbf{0}, \boldsymbol{I})$, and $\boldsymbol{\theta}*$ denotes the parameters in an unconditional score model $\boldsymbol{s}_{\boldsymbol{\theta}*}(\mathbf{x}, t)$. Here the iteration function $\boldsymbol{h}$ takes a noisy sample $\hat{\mathbf{x}}_{t_i}$ and reduces the noise therein to generate $\hat{\mathbf{x}}_{t_{i-1}}$, using the unconditional score model $\boldsymbol{s}_{\boldsymbol{\theta}*}(\mathbf{x}, t)$. For example, for the Euler-Maruyama sampler detailed in Algorithm 1, this iteration function is given by

$$\boldsymbol{h}(\hat{\mathbf{x}}_{t_i}, \mathbf{z}_i, \boldsymbol{s}_{\boldsymbol{\theta}*}(\hat{\mathbf{x}}_{t_i}, t_i)) = \hat{\mathbf{x}}_{t_i} - f(t_i)\hat{\mathbf{x}}_{t_i}/N + g(t_i)^2 \boldsymbol{s}_{\boldsymbol{\theta}*}(\hat{\mathbf{x}}_{t_i}, t_i)/N + g(t_i)\mathbf{z}_i/\sqrt{N}.$$

Samples obtained by this procedure $\{\hat{\mathbf{x}}_{t_i}\}_{i=0}^N$ constitute an approximation of $\{\mathbf{x}_t\}_{t \in [0,1]}$, where the last sample $\hat{\mathbf{x}}_{t_0}$ can be viewed as an approximate sample from $p_0(\mathbf{x})$. Most existing sampling

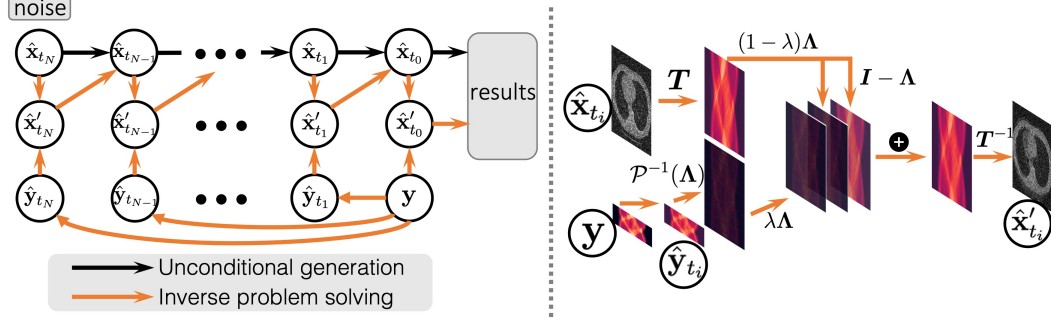

Figure 3: *(Left)* An overview of our method for solving inverse problems with score-based generative models. *(Right)* An illustration about how to combine $\hat{\mathbf{x}}_{t_i}$ and $\mathbf{y}$ to form $\hat{\mathbf{x}}'_{t_i}$.

methods for score-based generative models are instances of this iterative sampling paradigm, including Algorithm 1, ALD (Song & Ermon, 2019), probability flow ODEs (Song et al., 2021) and Predictor-Corrector samplers (Song et al., 2021).

To enforce the constraint implied by $\{\mathbf{y}_t \mid \mathbf{y}\}_{t \in [0,1]}$, we prepend an additional step to the iteration rule in Eq. (5), leading to

$$\hat{\mathbf{x}}'_{t_i} = \boldsymbol{k}(\hat{\mathbf{x}}_{t_i}, \hat{\mathbf{y}}_{t_i}, \lambda) \tag{6}$$

$$\hat{\mathbf{x}}_{t_{i-1}} = \boldsymbol{h}(\hat{\mathbf{x}}'_{t_i}, \mathbf{z}_i, \boldsymbol{s}_{\boldsymbol{\theta}*}(\hat{\mathbf{x}}_{t_i}, t_i)), \quad i = N, N-1, \cdots, 1, \tag{7}$$

where $\hat{\mathbf{x}}_{t_N} \sim \pi(\mathbf{x})$, $\hat{\mathbf{y}}_{t_i} \sim p_{t_i}(\mathbf{y}_{t_i} \mid \mathbf{y})$, and $0 \leqslant \lambda \leqslant 1$ is a hyper-parameter. We provide an illustration of this process in Fig. 3. The iteration function $\boldsymbol{k}(\cdot, \hat{\mathbf{y}}_{t_i}, \lambda) : \mathbb{R}^n \to \mathbb{R}^n$ promotes data consistency by solving a proximal optimization step (Nesterov, 2003; Boyd et al., 2004; Hammernik et al., 2021) that simultaneously minimizes the distance between $\hat{\mathbf{x}}'_{t_i}$ and $\hat{\mathbf{x}}_{t_i}$, and the distance between $\hat{\mathbf{x}}'_{t_i}$ and the hyperplane $\{\boldsymbol{x} \in \mathbb{R}^n \mid \boldsymbol{A}\boldsymbol{x} = \hat{\mathbf{y}}_{t_i}\}$, with a hyperparameter $0 \leqslant \lambda \leqslant 1$ balancing between the two:

$$\hat{\mathbf{x}}'_{t_i} = \underset{\boldsymbol{z} \in \mathbb{R}^n}{\arg\min}\{(1-\lambda)\|\boldsymbol{z} - \hat{\mathbf{x}}_{t_i}\|_{\boldsymbol{T}}^2 + \min_{\boldsymbol{u} \in \mathbb{R}^n} \lambda \|\boldsymbol{z} - \boldsymbol{u}\|_{\boldsymbol{T}}^2\} \quad s.t. \quad \boldsymbol{A}\boldsymbol{u} = \hat{\mathbf{y}}_{t_i}. \tag{8}$$

Recall that $\boldsymbol{A} = \mathcal{P}(\boldsymbol{\Lambda})\boldsymbol{T}$ according to Proposition 1. In the equation above we choose the norm $\|\boldsymbol{a}\|_{\boldsymbol{T}}^2 := \|\boldsymbol{T}\boldsymbol{a}\|_2^2$ to simplify our theoretical analysis. The decomposition in Proposition 1 allows us to derive a closed-form solution to the optimization problem in Eq. (8), as given below:

**Theorem 1.** *The solution of Eq. (8) can be given by*

$$\hat{\mathbf{x}}'_{t_i} = \boldsymbol{T}^{-1}[\lambda\boldsymbol{\Lambda}\mathcal{P}^{-1}(\boldsymbol{\Lambda})\hat{\mathbf{y}}_{t_i} + (1-\lambda)\boldsymbol{\Lambda}\boldsymbol{T}\hat{\mathbf{x}}_{t_i} + (\boldsymbol{I} - \boldsymbol{\Lambda})\boldsymbol{T}\hat{\mathbf{x}}_{t_i}], \tag{9}$$

*where $\mathcal{P}^{-1}(\boldsymbol{\Lambda}) : \mathbb{R}^m \to \mathbb{R}^n$ denotes any right inverse of $\mathcal{P}(\boldsymbol{\Lambda})$.*

See Fig. 3 for an illustration of the function $\hat{\mathbf{x}}'_{t_i} = \boldsymbol{k}(\hat{\mathbf{x}}_{t_i}, \hat{\mathbf{y}}_{t_i}, \lambda)$. The right inverse $\mathcal{P}^{-1}(\boldsymbol{\Lambda})$ increases the dimensionality of a vector $\boldsymbol{a} \in \mathbb{R}^m$ to $n$ by putting its entries on every index $i$ of an $n$-dimensional vector where $\boldsymbol{\Lambda}_{ii} = 1$. Recall that in sparse-view CT or undersampled MRI, $\text{diag}(\boldsymbol{\Lambda})$ represents a subsampling mask, and $\mathcal{P}(\boldsymbol{\Lambda})$ subsamples the full sinogram/k-space to generate the observation $\mathbf{y}$. In this case, $\mathcal{P}^{-1}(\boldsymbol{\Lambda})$ pads the observation $\mathbf{y}$ so that it has the same size as the full sinogram/k-space.

When $\lambda = 0$, $\hat{\mathbf{x}}'_{t_i} = \boldsymbol{k}(\hat{\mathbf{x}}_{t_i}, \hat{\mathbf{y}}_{t_i}, 0) = \hat{\mathbf{x}}_{t_i}$ completely ignores the constraint $\boldsymbol{A}\hat{\mathbf{x}}'_{t_i} = \hat{\mathbf{y}}_{t_i}$, in which case our sampling method in Eq. (7) performs unconditional generation. On the other hand, when $\lambda = 1$, $\hat{\mathbf{x}}'_{t_i} = \boldsymbol{k}(\hat{\mathbf{x}}_{t_i}, \hat{\mathbf{y}}_{t_i}, 1)$ satisfies $\boldsymbol{A}\hat{\mathbf{x}}'_{t_i} = \hat{\mathbf{y}}_{t_i}$ exactly. When the measurement is noisy, we choose $0 < \lambda < 1$ to allow slackness in the constraint $\boldsymbol{A}\hat{\mathbf{x}}'_{t_i} = \hat{\mathbf{y}}_{t_i}$. The value of $\lambda$ is important for balancing between $\hat{\mathbf{x}}'_{t_i} \approx \hat{\mathbf{x}}_{t_i}$ and $\boldsymbol{A}\hat{\mathbf{x}}'_{t_i} \approx \hat{\mathbf{y}}_{t_i}$. In practice, we use Bayesian optimization to tune this $\lambda$ automatically on a validation dataset. When the measurement process contains no noise, we replace $\hat{\mathbf{x}}_{t_0}$ with $\boldsymbol{k}(\hat{\mathbf{x}}_{t_0}, \mathbf{y}, 1)$ at the last sampling step to guarantee $\boldsymbol{A}\hat{\mathbf{x}}_{t_0} = \mathbf{y}$.

In summary, our method given in Eq. (7) introduces minimal modifications to an existing iterative sampling method of score-based generative models. For example, we can convert the sampler in Algorithm 1 to an inverse problem solver in Algorithm 2 by adding/modifying just three lines of

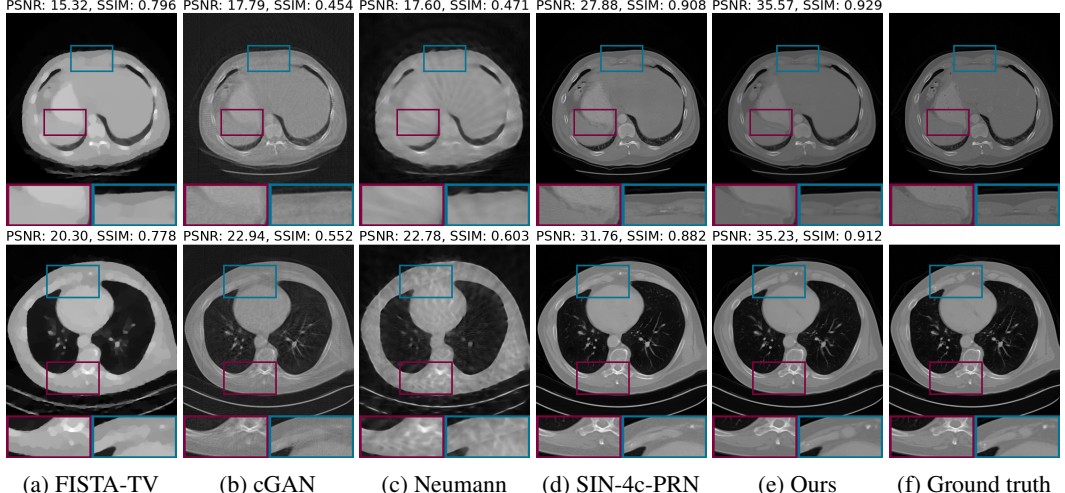

PSNR: 15.32, SSIM: 0.796  PSNR: 17.79, SSIM: 0.454  PSNR: 17.60, SSIM: 0.471  PSNR: 27.88, SSIM: 0.908  PSNR: 35.57, SSIM: 0.929

PSNR: 20.30, SSIM: 0.778  PSNR: 22.94, SSIM: 0.552  PSNR: 22.78, SSIM: 0.603  PSNR: 31.76, SSIM: 0.882  PSNR: 35.23, SSIM: 0.912

(a) FISTA-TV     (b) cGAN     (c) Neumann     (d) SIN-4c-PRN     (e) Ours     (f) Ground truth

Figure 4: Examples of sparse-view CT reconstruction results on LIDC $320 \times 320$ *(Top row)* and LDCT $512 \times 512$ *(Bottom row)*, all with 23 projections. You may zoom in to view more details.

pseudo-code. Unlike the concurrent work Jalal et al. (2021), our method is not limited to annealed Langevin dynamics (ALD). As demonstrated in our experiments, we outperform Jalal et al. (2021) even with the same ALD sampler, and can widen the performance gap further by using more advanced approaches like the Predictor-Corrector sampler (Song et al., 2021). Unlike Kadkhodaie & Simoncelli (2020); Kawar et al. (2021), we rely on the efficient alternative representation of $A$ given in Section 3.1, and do not require expensive SVD computation.

## 4 EXPERIMENTS

We aim to answer the following questions in this section: (1) Can we directly compete with best-in-class supervised learning techniques for the same measurement process used in their training, even though our approach is fully unsupervised? (2) Can our method generalize better to new measurement processes? (3) How do we fare against other unsupervised approaches? To study these questions, we experiment on several tasks in medical imaging, including sparse-view CT reconstruction, metal artifact removal (MAR) for CT, and undersampled MRI reconstruction. More experimental details are provided in Appendix B.

**Datasets** We consider two datasets for CT experiments. The first is the Lung Image Database Consortium (LIDC) image collection dataset (Armato III et al., 2011; Clark et al., 2013) where we slice the original 3D CT volumes to obtain 130304 2D images of resolution $320 \times 320$ for training. The second is the Low Dose CT (LDCT) Image and Projection dataset (Moen et al., 2021) that contains CT scans of multiple anatomic sites, including head, chest, and abdomen, from which we generate 47006 2D image slices of resolution $512 \times 512$ for training. We simulate CT measurements (sinograms) with a parallel-beam geometry using projection angles equally distributed across 180 degrees. For MAR experiments, we follow Yu et al. (2020) to synthesize metal artifacts. For undersampled MRI experiments, we use the Brain Tumor Segmentation (BraTS) 2021 dataset (Menze et al., 2014; Bakas et al., 2017), where we slice 3D MRI volumes to get 297270 images of resolution $240 \times 240$ as the training dataset. We simulate MRI measurements with Fast Fourier Transform using a single-coil setup, and follow Zbontar et al. (2018); Knoll et al. (2020) to undersample the k-space with an equispaced Cartesian mask. The performance is measured on 1000 test images with peak signal-to-noise ratio (PSNR) and structural similarity (SSIM).

**Standard techniques in medical imaging** We include two standard learning-free techniques as baselines for sparse-view CT reconstruction. The first is filtered back projection on sparse-view sinograms, which is denoted by "FBP". The second is an iterative reconstruction method with total variation regularization called FISTA-TV (Beck & Teboulle, 2009). For MAR experiments, we include another learning-free baseline called linear interpolation (LI, Kalender et al., 1987).

Table 1: Results for undersampled MRI reconstruction on BraTS. First two methods are supervised learning techniques trained with $8\times$ acceleration. The others are unsupervised techniques.

| Method | $24\times$ Acceleration | | $8\times$ Acceleration | | $4\times$ Acceleration | |
|---|---|---|---|---|---|---|
| | PSNR↑ | SSIM↑ | PSNR↑ | SSIM↑ | PSNR↑ | SSIM↑ |
| Cascade DenseNet | $23.39_{\pm2.17}$ | $0.765_{\pm0.042}$ | $28.35_{\pm2.30}$ | $0.845_{\pm0.038}$ | $30.97_{\pm2.33}$ | $0.902_{\pm0.028}$ |
| DuDoRNet | $18.46_{\pm3.05}$ | $0.662_{\pm0.093}$ | $\mathbf{37.88_{\pm3.03}}$ | $\mathbf{0.985_{\pm0.007}}$ | $30.53_{\pm4.13}$ | $0.891_{\pm0.071}$ |
| Score SDE | $27.83_{\pm2.73}$ | $0.849_{\pm0.038}$ | $35.04_{\pm2.11}$ | $0.943_{\pm0.016}$ | $37.55_{\pm2.08}$ | $0.960_{\pm0.013}$ |
| Langevin | $28.80_{\pm3.21}$ | $0.873_{\pm0.039}$ | $36.44_{\pm2.28}$ | $0.952_{\pm0.016}$ | $38.76_{\pm2.32}$ | $0.966_{\pm0.012}$ |
| Ours | $\mathbf{29.42_{\pm3.03}}$ | $\mathbf{0.880_{\pm0.035}}$ | $37.63_{\pm2.70}$ | $0.958_{\pm0.015}$ | $\mathbf{39.91_{\pm2.67}}$ | $\mathbf{0.965_{\pm0.013}}$ |

Table 2: Results for sparse-view CT reconstruction on LIDC and LDCT. FISTA-TV is a standard iterative reconstruction method that does not need training. cGAN, Neumann, and SIN-4c-PRN are supervised learning techniques trained with 23 projection angles.

| Method | Projections | LIDC $320 \times 320$ | | LDCT $512 \times 512$ | |
|---|---|---|---|---|---|
| | | PSNR↑ | SSIM↑ | PSNR↑ | SSIM↑ |
| FBP | 23 | $10.18_{\pm1.38}$ | $0.230_{\pm0.072}$ | $10.11_{\pm1.19}$ | $0.302_{\pm0.078}$ |
| FISTA-TV | 23 | $20.08_{\pm4.89}$ | $0.799_{\pm0.061}$ | $21.88_{\pm4.42}$ | $0.850_{\pm0.067}$ |
| cGAN | 23 | $19.83_{\pm3.07}$ | $0.479_{\pm0.103}$ | $19.90_{\pm2.52}$ | $0.545_{\pm0.065}$ |
| Neumann | 23 | $17.18_{\pm3.79}$ | $0.454_{\pm0.128}$ | $18.83_{\pm3.29}$ | $0.525_{\pm0.073}$ |
| SIN-4c-PRN | 23 | $30.48_{\pm3.99}$ | $0.895_{\pm0.047}$ | $34.82_{\pm3.55}$ | $0.877_{\pm0.116}$ |
| | 10 | $29.52_{\pm2.63}$ | $0.823_{\pm0.061}$ | $28.96_{\pm4.41}$ | $0.849_{\pm0.086}$ |
| Ours | 20 | $34.40_{\pm2.66}$ | $0.895_{\pm0.048}$ | $36.80_{\pm4.50}$ | $0.936_{\pm0.058}$ |
| | 23 | $\mathbf{35.24_{\pm2.71}}$ | $\mathbf{0.905_{\pm0.046}}$ | $\mathbf{37.41_{\pm4.62}}$ | $\mathbf{0.941_{\pm0.057}}$ |

**Supervised learning baselines** For sparse-view CT on both LIDC and LDCT, we include cGAN (Ghani & Karl, 2018), Neumann (Gilton et al., 2019), and SIN-4c-PRN (Wei et al., 2020) as supervised learning baselines. We follow the settings in Wei et al. (2020) and train all methods with 23 projection angles. For MAR, we use cGANMAR (Wang et al., 2018) and SNMAR (Yu et al., 2020) as the baselines. For undersampled MRI on BraTS, we compare against Cascade DenseNet (Zheng et al., 2019) and DuDoRNet (Zhou & Zhou, 2020), which are both trained with a $8\times$ acceleration factor by measuring only $1/8$ of the full k-space.

**Unsupervised learning baselines** For unsupervised techniques, so far only score-based generative models have witnessed success on clinic data. We compare with several existing methods that apply score-based generative models to inverse problem solving. Specifically, we consider the "Langevin" approach proposed in Jalal et al. (2021), and the "Score SDE" method in Song et al. (2021), where the former is limited to annealed Langevin dynamics (ALD) sampling, and the latter was based on a crude approximation to the conditional score function $\nabla_{\mathbf{x}_t} \log p_t(\mathbf{x}_t \mid \mathbf{y})$ in Eq. (4), and was proposed as a theoretical possibility in Appendix I.4 of Song et al. (2021) without experiments. We only focus on undersampled MRI for these baselines, since it is the only medical imaging problem ever tackled with score-based generative models before our work. All methods share the same score models and only differ in terms of inference. We make sure all sampling algorithms have comparable number of iteration steps ($N$ in Eqs. (5) and (7)).

**Competing with supervised learning approaches** Thanks to the outstanding sample quality of score-based generative models, we can achieve comparable or better performance than best-in-class supervised learning methods even for the same measurement process used in their training. As shown in Table 2, we outperform the top supervised learning technique SIN-4c-PRN on sparse-view CT reconstruction by a significant margin, on both the LIDC and LDCT datasets. Our results with

Table 3: MAR results on LIDC.

| Method | PSNR↑ | SSIM↑ |
|---|---|---|
| LI | $26.30_{\pm2.62}$ | $0.910_{\pm0.028}$ |
| cGANMAR | $27.27_{\pm1.96}$ | $0.927_{\pm0.060}$ |
| SNMAR | $27.28_{\pm1.43}$ | $0.937_{\pm0.048}$ |
| Ours | $\mathbf{32.16_{\pm2.32}}$ | $\mathbf{0.939_{\pm0.022}}$ |

20 measurements are even better than supervised learning counterparts with 23 measurements. In Fig. 4, we provide a visual comparison of the reconstruction quality for various methods, where it is clear to see that our method can recover more details faithfully. From results in Table 3, we also outperform the top supervised learning method SNMAR on metal artifact removal. As shown

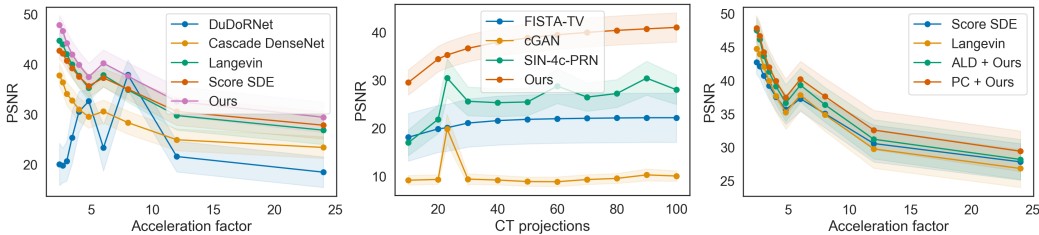

Figure 5: Performance vs. numbers of measurements. Shaded areas represent standard deviation. *(Left)* MRI on BraTS. *(Center)* CT on LIDC. *(Right)* Comparing score-based generative models for undersampled MRI reconstruction on BraTS.

in Fig. 7, our method generates images with less artifacts and preserves the structure better. For undersampled MRI reconstruction results given in Tables 1 and 3, our method is ranked the 2nd for the case of $8\times$ acceleration, with comparable performance to the top supervised method DuDoRNet.

**Generalizing to different number of measurements** Since our approach is fully unsupervised, we can naturally apply the same score model to different measurement processes. We first consider changing the number of measurements at the test time, *e.g.*, using different number of projection angles (resp. different acceleration factors) for sparse-view CT (resp. undersampled MRI) reconstruction. As shown in Table 1 and Fig. 5 *(Left)*, we achieve the best performance on undersampled MRI for both $24\times$ and $4\times$ acceleration factors, whereas DuDoRNet fails to generalize when the acceleration factor changes. The other supervised learning approach Cascade DenseNet demonstrates limited adaptability by building a model architecture inspired by the physical measurement process of MRI, but fails to yield top-level performance. For sparse-view CT reconstruction, all supervised learning methods struggle to generalize to different projection angles, as shown in Fig. 5 *(Center)*.

**Generalizing to different measurement processes in CT** We can perform both sparse-view CT reconstruction and metal artifact removal (MAR) with a single score model trained on CT images. These two tasks are inverse problems in CT imaging with different measurement processes $A$, but they share the same $T$ in the decomposition of Proposition 1. We provide a visualization of the measurement process corresponding to MAR in Fig. 8. As shown in Table 3, we can outperform supervised learning techniques specifically designed and trained for MAR, while using the same score model used in sparse-view CT reconstruction on LIDC.

**Comparing against existing score-based methods** We compare our method against Langevin (Jalal et al., 2021) and Score SDE (Song et al., 2021) for undersampled MRI reconstruction on BraTS. Two variants of our approach are considered, which respectively use annealed Langevin dynamics (ALD) and the Predictor-Corrector (PC) sampler for score-based generative models as the backend. We denote the former by "ALD + Ours", and the latter by "PC + Ours" (our default method for all other experiments). Recall that Langevin uses ALD as the sampler, same as "ALD + Ours". All results are provided in Fig. 5 *(Right)*. We observe that "ALD + Ours" uniformly outperform Langevin and Score SDE across all numbers of measurements in the experiment. Moreover, "PC + Ours" can further improve "ALD + Ours", demonstrating the power of switching to more advanced sampling methods of score-based generative models in our proposed approach.

# 5 CONCLUSION

We propose a new method to solve linear inverse problems with score-based generative models. Our method is fully unsupervised, requires no paired data for training, can flexibly adapt to different measurement processes at test time, and only requires minimal modifications to a large number of existing sampling methods of score-based generative models. Empirical results demonstrate that our method can match or outperform existing supervised learning counterparts on image reconstruction for sparse-view CT and undersampled MRI, and has better generalization to new measurement processes, such as using a different number of projections or downsampling ratios in CT/MRI, and tackling both sparse-view CT reconstruction and metal artifact removal with a single model.

AUTHOR CONTRIBUTIONS

Yang Song designed the project, wrote the paper, and ran all experiments for score-based generative models. Liyue Shen preprocessed data, ran all baseline experiments, and helped write the paper. Lei Xing and Stefano Ermon supervised the project, provided valuable feedback, and helped edit the paper.

ACKNOWLEDGMENTS

YS is supported by the Apple PhD Fellowship in AI/ML. LS is supported by the Stanford Bio-X Graduate Student Fellowship. This research was supported by NSF (#1651565, #1522054, #1733686), ONR (N000141912145), AFOSR (FA95501910024), ARO (W911NF-21-1-0125), Sloan Fellowship, and Google TPU Research Cloud. This research was also supported by NIH/NCI (1R01 CA256890 and 1R01 CA227713).

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

## A    PROOFS

**Proposition 1.** *If* $\mathrm{rank}(\boldsymbol{A}) = m$, *then there exist an invertible matrix* $\boldsymbol{T} \in \mathbb{R}^{n \times n}$, *and a diagonal matrix* $\boldsymbol{\Lambda} \in \{0,1\}^{n \times n}$ *with* $\mathrm{tr}(\boldsymbol{\Lambda}) = m$, *such that* $\boldsymbol{A} = \mathcal{P}(\boldsymbol{\Lambda})\boldsymbol{T}$. *Here* $\mathcal{P}(\boldsymbol{\Lambda}) \in \{0,1\}^{m \times n}$ *is an operator that, when multiplied with any vector* $\boldsymbol{a} \in \mathbb{R}^n$, *reduces its dimensionality to* $m$ *by removing each* $i$-*th element of* $\boldsymbol{a}$ *for* $i = 1, 2, \cdots, n$ *if* $\boldsymbol{\Lambda}_{ii} = 0$.

*Proof.* Let $\boldsymbol{A} = (\boldsymbol{a}_1^{\mathsf{T}}, \boldsymbol{a}_2^{\mathsf{T}}, \cdots, \boldsymbol{a}_m^{\mathsf{T}}) \in \mathbb{R}^{m \times n}$.  Since $\boldsymbol{A}$ has full rank, the row vectors $\{\boldsymbol{a}_1, \boldsymbol{a}_2, \cdots, \boldsymbol{a}_m\}$ are linearly independent. We can therefore extend them to a total of $n$ linearly independent vectors, *i.e.*, $\{\boldsymbol{a}_1, \boldsymbol{a}_2, \cdots, \boldsymbol{a}_m, \boldsymbol{b}_1, \cdots, \boldsymbol{b}_{n-m}\}$. Due to the linear independence, we know $\boldsymbol{T} = (\boldsymbol{a}_1^{\mathsf{T}}, \boldsymbol{a}_2^{\mathsf{T}}, \cdots, \boldsymbol{a}_m^{\mathsf{T}}, \boldsymbol{b}_1^{\mathsf{T}}, \cdots, \boldsymbol{b}_{n-m}^{\mathsf{T}}) \in \mathbb{R}^{n \times n}$ has full rank and is invertible. Next, we define

$$\boldsymbol{\Lambda} = \mathrm{diag}(\underbrace{1, 1, \cdots, 1}_{m}, \underbrace{0, 0, \cdots, 0}_{n-m}),$$

where $\mathrm{diag}$ converts a vector to a diagonal matrix. Clearly $\mathrm{tr}(\boldsymbol{\Lambda}) = m$ and $\boldsymbol{A} = \mathcal{P}(\boldsymbol{\Lambda})\boldsymbol{T}$, which completes the proof. $\qquad\square$

**Lemma 1.** *Let* $\mathcal{P}^{-1}(\boldsymbol{\Lambda}) : \mathbb{R}^m \to \mathbb{R}^n$ *be any right inverse of* $\mathcal{P}(\boldsymbol{\Lambda}) : \mathbb{R}^n \to \mathbb{R}^m$. *For any* $\boldsymbol{u} \in \mathbb{R}^n$ *and* $\hat{\mathbf{y}}_t \in \mathbb{R}^m$, *we have*

$$\mathcal{P}(\boldsymbol{\Lambda})\boldsymbol{T}\boldsymbol{u} = \hat{\mathbf{y}}_t \iff \boldsymbol{\Lambda}\boldsymbol{T}\boldsymbol{u} = \boldsymbol{\Lambda}\mathcal{P}^{-1}(\boldsymbol{\Lambda})\hat{\mathbf{y}}_t$$

*Proof.* By the definition of $\mathcal{P}(\boldsymbol{\Lambda})$, we have $\mathcal{P}(\boldsymbol{\Lambda}) = \mathcal{P}(\boldsymbol{\Lambda})\boldsymbol{\Lambda}$, and

$$\forall \boldsymbol{a} \in \mathbb{R}^n, \boldsymbol{b} \in \mathbb{R}^n : \quad \mathcal{P}(\boldsymbol{\Lambda})\boldsymbol{a} = \mathcal{P}(\boldsymbol{\Lambda})\boldsymbol{b} \iff \boldsymbol{\Lambda}\boldsymbol{a} = \boldsymbol{\Lambda}\boldsymbol{b}. \tag{10}$$

To prove the "if" direction, we note that

$$\begin{aligned}
\boldsymbol{\Lambda}\boldsymbol{T}\boldsymbol{u} = \boldsymbol{\Lambda}\mathcal{P}^{-1}(\boldsymbol{\Lambda})\hat{\mathbf{y}}_t &\implies \mathcal{P}(\boldsymbol{\Lambda})\boldsymbol{\Lambda}\boldsymbol{T}\boldsymbol{u} = \mathcal{P}(\boldsymbol{\Lambda})\boldsymbol{\Lambda}\mathcal{P}^{-1}(\boldsymbol{\Lambda})\hat{\mathbf{y}}_t \\
&\implies \mathcal{P}(\boldsymbol{\Lambda})\boldsymbol{T}\boldsymbol{u} = \mathcal{P}(\boldsymbol{\Lambda})\mathcal{P}^{-1}(\boldsymbol{\Lambda})\hat{\mathbf{y}}_t \\
&\implies \mathcal{P}(\boldsymbol{\Lambda})\boldsymbol{T}\boldsymbol{u} = \hat{\mathbf{y}}_t.
\end{aligned}$$

To prove the "only if" direction, we have

$$\begin{aligned}
\mathcal{P}(\boldsymbol{\Lambda})\boldsymbol{T}\boldsymbol{u} = \hat{\mathbf{y}}_t &\implies \mathcal{P}(\boldsymbol{\Lambda})\boldsymbol{T}\boldsymbol{u} = \mathcal{P}(\boldsymbol{\Lambda})\mathcal{P}^{-1}(\boldsymbol{\Lambda})\hat{\mathbf{y}}_t \\
&\overset{(i)}{\implies} \boldsymbol{\Lambda}\boldsymbol{T}\boldsymbol{u} = \boldsymbol{\Lambda}\mathcal{P}^{-1}(\boldsymbol{\Lambda})\hat{\mathbf{y}}_t,
\end{aligned}$$

where (i) is due to the property in Eq. (10). This completes the proof for both directions.

$\qquad\square$

**Theorem 1.** *The solution of Eq.* (8) *can be given by*

$$\hat{\mathbf{x}}'_{t_i} = \boldsymbol{T}^{-1}[\lambda\boldsymbol{\Lambda}\mathcal{P}^{-1}(\boldsymbol{\Lambda})\hat{\mathbf{y}}_{t_i} + (1 - \lambda)\boldsymbol{\Lambda}\boldsymbol{T}\hat{\mathbf{x}}_{t_i} + (\boldsymbol{I} - \boldsymbol{\Lambda})\boldsymbol{T}\hat{\mathbf{x}}_{t_i}], \tag{9}$$

*where* $\mathcal{P}^{-1}(\boldsymbol{\Lambda}) : \mathbb{R}^m \to \mathbb{R}^n$ *denotes any right inverse of* $\mathcal{P}(\boldsymbol{\Lambda})$.

*Proof.* The optimization objective function in Eq. (8) can be written as

$$\begin{aligned}
&(1 - \lambda)\|\boldsymbol{z} - \hat{\mathbf{x}}_t\|_{\boldsymbol{T}}^2 + \lambda\|\boldsymbol{z} - \boldsymbol{u}\|_{\boldsymbol{T}}^2 \\
=&(1 - \lambda)\|\boldsymbol{T}\boldsymbol{z} - \boldsymbol{T}\hat{\mathbf{x}}_t\|_2^2 + \lambda\|\boldsymbol{T}\boldsymbol{z} - \boldsymbol{T}\boldsymbol{u}\|_2^2 \\
=&(1 - \lambda)\|\boldsymbol{T}\boldsymbol{z} - \boldsymbol{T}\hat{\mathbf{x}}_t\|_2^2 + \lambda\|\boldsymbol{\Lambda}\boldsymbol{T}(\boldsymbol{z} - \boldsymbol{u}) + (\boldsymbol{I} - \boldsymbol{\Lambda})\boldsymbol{T}(\boldsymbol{z} - \boldsymbol{u})\|_2^2 \\
=&(1 - \lambda)\|\boldsymbol{T}\boldsymbol{z} - \boldsymbol{T}\hat{\mathbf{x}}_t\|_2^2 + \lambda\|\boldsymbol{\Lambda}\boldsymbol{T}(\boldsymbol{z} - \boldsymbol{u})\|_2^2 + \lambda\|(\boldsymbol{I} - \boldsymbol{\Lambda})\boldsymbol{T}(\boldsymbol{z} - \boldsymbol{u})\|_2^2 \\
=&(1 - \lambda)\|\boldsymbol{T}\boldsymbol{z} - \boldsymbol{T}\hat{\mathbf{x}}_t\|_2^2 + \lambda\|\boldsymbol{\Lambda}\boldsymbol{T}\boldsymbol{z} - \boldsymbol{\Lambda}\mathcal{P}^{-1}(\boldsymbol{\Lambda})\hat{\mathbf{y}}_t\|_2^2 + \lambda\|(\boldsymbol{I} - \boldsymbol{\Lambda})\boldsymbol{T}(\boldsymbol{z} - \boldsymbol{u})\|_2^2
\end{aligned}$$

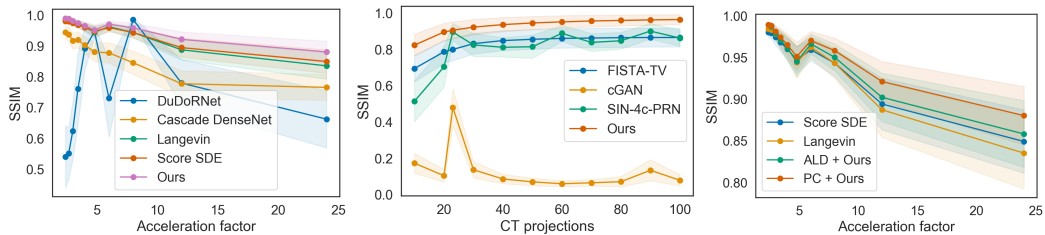

Figure 6: SSIM vs. numbers of measurements. Shaded areas represent standard deviation. *(Left)* MRI on BraTS. *(Center)* CT on LIDC. *(Right)* Comparing score-based generative models for undersampled MRI reconstruction on BraTS.

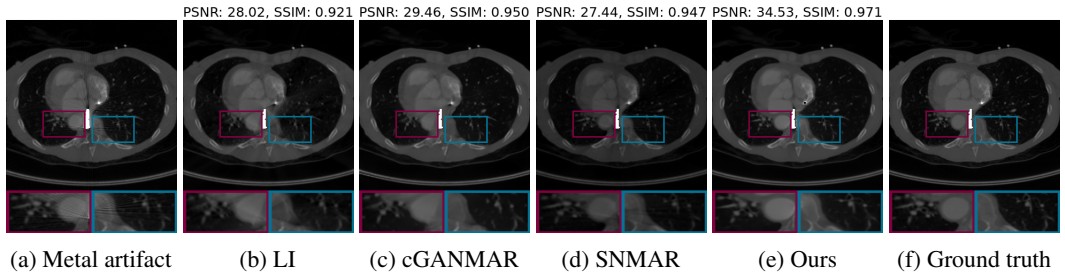

(a) Metal artifact    (b) LI    (c) cGANMAR    (d) SNMAR    (e) Ours    (f) Ground truth

Figure 7: Examples of metal artifact removal on LIDC. You may zoom in to view more details.

Since $\boldsymbol{Au} = \hat{\mathbf{y}}_t$, we have $\mathcal{P}(\boldsymbol{\Lambda})\boldsymbol{Tu} = \hat{\mathbf{y}}_t$ and equivalently $\boldsymbol{\Lambda Tu} = \boldsymbol{\Lambda}\mathcal{P}^{-1}(\boldsymbol{\Lambda})\hat{\mathbf{y}}_t$ due to Lemma 1. This constraint does not restrict the value of $(\boldsymbol{I} - \boldsymbol{\Lambda})\boldsymbol{Tu}$. Therefore, when $\boldsymbol{Au} = \hat{\mathbf{y}}_t$, we have

$$\|\boldsymbol{z} - \hat{\mathbf{x}}_t\|_{\boldsymbol{T}}^2 + \min_{\boldsymbol{u}}(1-\lambda)\lambda \|\boldsymbol{z} - \boldsymbol{u}\|_{\boldsymbol{T}}^2$$

$$=(1-\lambda)\|\boldsymbol{Tz} - \boldsymbol{T}\hat{\mathbf{x}}_t\|_2^2 + \min_{\boldsymbol{u}} \lambda \left\|\boldsymbol{\Lambda Tz} - \boldsymbol{\Lambda}\mathcal{P}^{-1}(\boldsymbol{\Lambda})\hat{\mathbf{y}}_t\right\|_2^2 + \lambda \|(\boldsymbol{I} - \boldsymbol{\Lambda})\boldsymbol{T}(\boldsymbol{z} - \boldsymbol{u})\|_2^2$$

$$=(1-\lambda)\|\boldsymbol{Tz} - \boldsymbol{T}\hat{\mathbf{x}}_t\|_2^2 + \lambda \left\|\boldsymbol{\Lambda Tz} - \boldsymbol{\Lambda}\mathcal{P}^{-1}(\boldsymbol{\Lambda})\hat{\mathbf{y}}_t\right\|_2^2$$

$$=(1-\lambda)\|\boldsymbol{\Lambda Tz} - \boldsymbol{\Lambda T}\hat{\mathbf{x}}_t\|_2^2 + \lambda \left\|\boldsymbol{\Lambda Tz} - \boldsymbol{\Lambda}\mathcal{P}^{-1}(\boldsymbol{\Lambda})\hat{\mathbf{y}}_t\right\|_2^2 + (1-\lambda)\|(\boldsymbol{I} - \boldsymbol{\Lambda})\boldsymbol{Tz} - (\boldsymbol{I} - \boldsymbol{\Lambda})\boldsymbol{T}\hat{\mathbf{x}}_t\|_2^2.$$

This simplifies the optimization problem in Eq. (8) to

$$\min_{\boldsymbol{z}}(1-\lambda)\|\boldsymbol{\Lambda Tz} - \boldsymbol{\Lambda T}\hat{\mathbf{x}}_t\|_2^2 + \lambda \left\|\boldsymbol{\Lambda Tz} - \boldsymbol{\Lambda}\mathcal{P}^{-1}(\boldsymbol{\Lambda})\hat{\mathbf{y}}_t\right\|_2^2 + (1-\lambda)\|(\boldsymbol{I} - \boldsymbol{\Lambda})\boldsymbol{Tz} - (\boldsymbol{I} - \boldsymbol{\Lambda})\boldsymbol{T}\hat{\mathbf{x}}_t\|_2^2,$$

which is minimizing a quadratic function of $\boldsymbol{z}$. The optimal solution $\boldsymbol{z}^*$ is thus in closed form:

$$\boldsymbol{z}^* = \boldsymbol{T}^{-1}[(\boldsymbol{I} - \boldsymbol{\Lambda})\boldsymbol{T}\hat{\mathbf{x}}_t + (1-\lambda)\boldsymbol{\Lambda T}\hat{\mathbf{x}}_t + \lambda\boldsymbol{\Lambda}\mathcal{P}^{-1}(\boldsymbol{\Lambda})\hat{\mathbf{y}}_t].$$

According to the definition, $\hat{\mathbf{x}}_t' = \boldsymbol{z}^*$, whereby the proof is completed. □

## B    ADDITIONAL EXPERIMENTAL DETAILS

### B.1    ADDITIONAL RESULTS

In Fig. 6, we provide SSIM results versus the number of measurements for multiple methods and tasks. In general, the SSIM curves have very similar trends to the PSNR curves in Fig. 5. We additionally provide a visualization of metal artifact removal results in Fig. 7.

### B.2    THE TASK OF METAL ARTIFACT REMOVAL

Metallic implants in an object can cause strong metal artifacts in CT imaging. As shown in Fig. 8, the source of artifacts come from extremely bright regions in the sinogram, called metal traces. To reduce or ideally remove metal artifacts from a CT image, we remove metal traces from the sinogram and

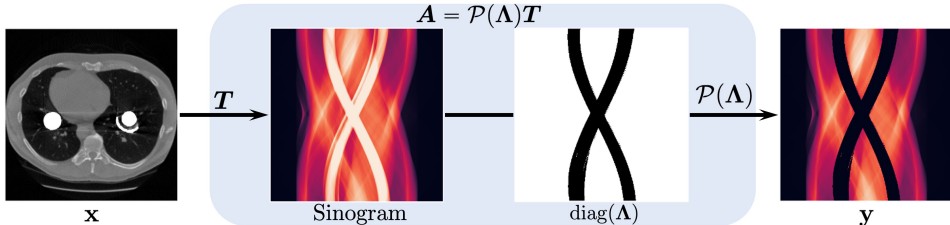

Figure 8: The linear measurement process of metal artifact removal.

leverage the data prior to complete the sinogram. As a result, metal artifact removal can be viewed as an inverse problem, where the measurement process gives the full sinogram except for the metal trace region, and our goal is to reconstruct the full CT image using this partially known sinogram, which will be artifact-free assuming perfect inpainting of the sinogram.

### B.3 DETAILS OF DATASETS

**CT datasets**  We conduct experiments of 2D CT image reconstruction on two datasets. First, the Lung Image Database Consortium image collection (LIDC) (Armato III et al., 2011; Clark et al., 2013) consists of diagnostic and lung cancer screening thoracic computed tomography (CT) scans for lung cancer detection and diagnosis, which contains 1018 cases. Second, the Low Dose CT Image and Projection dataset (LDCT) (Clark et al., 2013; Moen et al., 2021) involves CT images of multiple anatomic sites, including 99 head CT scans, 100 chest CT scans, and 100 abdomen CT scans. Note that for the LDCT dataset, we only use the full-dose CT images in our experiments. In CT image processing, we convert the Hounsfield units from dicom files to the attenuation coefficients and set the background pixels to zero. Then, 2D CT images are sliced from 3D CT volumes. The sinograms are simulated from 2D CT images based on parallel-beam geometry with different number of projection angles that are equally distributed across 180 degrees.

**MRI dataset**  The Brain Tumor Segmentation (BraTS) 2021 dataset (Menze et al., 2014; Bakas et al., 2017) collected for the image segmentation challenge contains 2000 cases (8000 MRI scans), where each case has four different MR contrasts: native (T1), post-contrast T1-weighted (T1Gd), T2-weighted (T2), and T2 Fluid Attenuated Inversion Recovery (T2-FLAIR). For each 3D MR volume, we extract 2D slices from 3D volumes and simulate k-space data by Fast Fourier Transform. To reconstruct MR images, we follow Knoll et al. (2020); Zbontar et al. (2018) to undersample k-space data with an equispaced Cartesian mask, where the center k-space is fully sampled while the left k-space is under-sampled by equispaced columns.

### B.4 DETAILS OF SCORE-BASED GENERATIVE MODELS

We use the NCSN++ model architecture in Song et al. (2021), and perturb the data with the Variance Exploding (VE) SDE. Our training procedure follows that of Song et al. (2021). Instead of generating samples according to the numerical SDE solver in Algorithm 1, we use the Predictor-Corrector (PC) sampler as described in Song et al. (2021) since it generally has better performance for VE SDEs. In PC samplers, the predictor refers to a numerical solver for the reverse-time SDE while the corrector can be any Markov chain Monte Carlo (MCMC) method that only depends on the scores. One such MCMC method considered in this work is Langevin dynamics, whereby we transform any initial sample $\mathbf{x}^{(0)}$ to an approximate sample from $p_t(\mathbf{x})$ via the following procedure:

$$\mathbf{x}^{(i+1)} \leftarrow \mathbf{x}^{(i)} + \epsilon \nabla_{\mathbf{x}} \log p_t(\mathbf{x}^{(i)}) + \sqrt{2\epsilon} \, \mathbf{z}^{(i)}, \quad i = 0, 1, \cdots, N-1. \tag{11}$$

Here $N \in \mathbb{N}_{>0}$, $\epsilon > 0$, and $\mathbf{z}^{(i)} \sim \mathcal{N}(\mathbf{0}, \mathbf{I})$. The theory of Langevin dynamics guarantees that in the limit of $N \to \infty$ and $\epsilon \to 0$, $\mathbf{x}^{(N)}$ is a sample from $p_t(\mathbf{x})$ under some regularity conditions. Note that Langevin dynamics only requires the knowledge of $\nabla_{\mathbf{x}} \log p_t(\mathbf{x})$, which can be approximated using the time-dependent score model $s_{\boldsymbol{\theta}*}(\mathbf{x}, t)$. In PC samplers, each predictor step immediately follows multiple consecutive corrector steps, all using the same $s_{\boldsymbol{\theta}*}(\mathbf{x}, t)$ evaluated at the same $t$. This jointly ensures that our intermediate sample at $t$ is approximately distributed according to $p_t(\mathbf{x})$. As shown

in Song et al. (2021), PC sampling often outperforms numerical solvers for the reverse-time SDE, especially when the forward SDE in Eq. (1) is a VE SDE. In order to use PC samplers for inverse problem solving, our modification is similar to the change made in Algorithm 2 for Algorithm 1. Specifically, we run line 4 & 5 in Algorithm 2 before every corrector or predictor step.

When comparing our approach to previous methods with score-based generative models, we use the same score model to isolate the confounding factors in model training and architecture design. Moreover, we make sure the total cost of sampling is comparable across different methods. For the ALD sampler used in Jalal et al. (2021), we use 700 noise scales with 3 steps of Langevin dynamics per noise scale, resulting in a total of $700 \times 3 = 2100$ steps that require score function evaluation. For the PC sampler, we use 1000 noise scales and 1 step of Langevin dynamics per noise scale, totalling $1000 + 1000 = 2000$ steps of score model evaluation.

For PC samplers, the step size $\epsilon$ in Langevin dynamics is determined by a signal-to-noise ratio $\eta$. For all methods, we tune $\eta$ and $\lambda$ in Eq. (8) with 100 steps of Bayesian optimization on a validation dataset, and report the results on the test dataset with the optimal parameters. We use the `ax-platform` toolkit for Bayesian optimization. The optimal parameters in our experiments are given by

- Sparse-view CT on LIDC $320 \times 320$: $\eta = 0.246$, $\lambda = 0.841$.
- Metal artifact removal on LIDC $320 \times 320$: $\eta = 0.209$, $\lambda = 0.227$.
- Sparse-view CT on LDCT $512 \times 512$: $\eta = 0.4$, $\lambda = 0.72$.
- Accelerated MRI on BraTS $240 \times 240$: $\eta = 0.577$, $\lambda = 0.982$.

### B.5 TRAINING DETAILS OF BASELINE MODELS

#### B.5.1 BASELINE MODELS FOR SPARSE-VIEW CT RECONSTRUCTION

**FBP** Filtered back projection (FBP) is a standard way for CT image reconstruction, which simply put the projections (sinogram) back to the image space based on the corresponding projection angles and geometry to get an approximated estimation of the unknown image. Usually, a high-pass filter, ramp filter is used to eliminate the blurring during this process. In our experiments, we conduct FBP on sparse-view sinograms using the torch radon toolbox (Ronchetti, 2020).

**FISTA-TV** FISTA-TV is a fast iterative shrinkage-thresholding algorithm (FISTA) for solving linear inverse problems in image processing (Beck & Teboulle, 2009). It adopts a total variation (TV) term as the regularization in the optimization procedure. Each optimization iteration involves a matrix-vector multiplication followed by a shrinkage-threshold step. In experiments, FISTA is implemented using the tomobar toolbox (Kazantsev & Wadeson, 2020) with the regularization using the CCPi regularisation toolkit (Kazantsev et al., 2019). We run 300 iterations for reconstructing each CT image with regularization parameter 0.001. Considering the nature of iterative reconstruction in FISTA, it is quite natural to generalize this method to different number of projections for reconstructing CT images. In experiments of generalizing to different number of measurements, FISTA method takes as input the sinogram with different numbers of projections and the corresponding angles for these input projections for the iterative procedure.

**cGAN** Conventional iterative CT reconstruction algorithms like FISTA are typically slow due to their iterative nature. Ghani & Karl (2018) proposed to cast sparse-view CT reconstruction as a sinogram inpainting problem. Specifically, it used a conditional generative adversarial network (cGAN) to first complete the sinogram data prior to reconstructing CT images, thereby avoiding the costly iterative tomographic processing. However, the imperfect sinogram inpainting may further cause image artifacts. Specifically, cGAN model takes zero-padded sparse-view sinogram with 23 projections as input and generates the completed full-angle sinogram with 180 projections. The cGAN model was implemented using PyTorch (Paszke et al., 2019) and trained using a batchsize of 64 and learning rate of 0.0001 with 50 epochs in total. In experiments of generalizing to different number of measurements, we deployed the trained cGAN model by zero-padding sparse-view sinogram with different numbers of projections to full-view sinogram as the input. After obtaining the output inpainted sinogram, we replace the corresponding projections in the output based on the ground truth projections in the input. Finally, the images were reconstructed from the overlayed sinogram. Note

that we trained the model using 23 projections and tested it on other projection settings to evaluate the generalization.

**SIN-4c-PRN**    To further reduce the artifacts in both sinogram and image space, SIN-4c-PRN (Wei et al., 2020) proposed a two-step sparse-view CT reconstruction model. It involves a sinogram inpainting network (SIN) to generate super-resolved sinograms with different number of projections, and then a post-processing refining network (PRN) to further remove image artifacts. Both networks are connected through a filtered back-projection operation (FBP). Specifically, SIN model takes 23-view sinogram as input to fistly upsample to full-view sinogram and then generate sinograms through network for 23, 45, 90, 180 projections respectively. FBP transforms these generated sinograms to image space, which was then concatenated and feed into PRN model for refinement. The framework was implemented using PyTorch (Paszke et al., 2019) while FBP operation was implemented using . SIN model was trained using a batchsize of 20 and learning rate of 0.0001, while PRN model was trained using a batchsize of 15 and learning rate of 0.0001. Considering that LIDC dataset is much larger than LDCT dataset, the SIN-4c-PRN model was trained for 30 epochs on LIDC dataset and 50 epochs on LDCT dataset. To deploy the trained SIN model to different numbers of measurements, the sinograms with various number of projections are taken as the input for SIN model to generate multi-view sinograms, which were also overlayed with corresponding ground truth projections in inputs. The generated multi-view sinograms are then used for PRN model inference. Since SIN-4c-PRN model involves the dual-domain learning in both sinogram and image spaces to remove artifacts, and generates multi-scale sinograms during sinogram inpainting, it shows a better generalization to different numbers of measurements compared with cGAN model as shown in Figure 5 and Figure 6.

**Neumann**    Meanwhile, in another parallel direction, researchers proposed to learn the regularizer used in optimization from training data, outperforming traditional regularizers. Specifically, Gilton et al. (2019) presented an end-to-end, data-driven method for learning a nonlinear regularizer for solving inverse problems inspired by the Neumann series, called Neumann network. Neumann network was implemented using PyTorch (Paszke et al., 2019). Due to GPU memory constraints, the model training used the batchsize of 5 on LIDC dataset and the batchsize of 2 on LDCT dataset. The initial learning rate was 0.00001 with an exponential learning rate decay. The network was trained with 15 training epochs on both datasets.

### B.5.2    BASELINE MODELS FOR UNDERSAMPLED MRI RECONSTRUCTION

**DuDoRNet**    Zhou & Zhou (2020) proposed a dual domain recurrent network (DuDoRNet) to simultaneously recover k-space data and images for MRI reconstruction, in order to address aliasing artifacts in both frequency and image domains. The original model in Zhou & Zhou (2020) also embedded a deep T1 prior to make use of fully-sampled short protocol (T1) as complementary information. For a fair comparison with other supervised learning approaches, in our experiments, we do not include this additional information but train the DuDoRNet model without T1 prior. The DuDoRNet was trained using a batchsize of 6 and a learning rate of 0.0005 with 5 training epochs. In experiments of generalizing to different number of measurements, we trained the model with an acceleration factor of 8 and deployed the trained model to other acceleration factors during testing. Specifically, for inference, we use different Cartesian masking function corresponding to different acceleration factors or down-sampling ratios to sub-sample the k-space data for the network input with the corresponding initial reconstructed image with zero-padding k-space.

**Cascade DenseNet**    To reconstruct de-aliased MR images from under-sampled k-space data, Zheng et al. (2019) proposed a cascaded dilated dense network (CDDN) for MRI reconstruction, based on stacked dense blocks with residual connections while using the zero-filled MR image as inputs. Specifically, they used a two-step data consistency layer for k-space correction, and replaced corresponding phase-coding lines of the generated image with the original sampled k-space data after each block. In experiments, we trained the model using a batchsize of 8 and a learning rate of 0.0001, with 5 epochs on BraTS dataset. In experiments of generalizing to different number of measurements, we trained the model with an acceleration factor of 8 and deployed the trained model to other acceleration factors during testing. Similarly, different masking functions corresponding to different acceleration factors were used to sub-sample k-space data to get network inputs. From results, we observe that

Cascaded DenseNet generalizes better to more measurements than DuDoRNet as shown in Figure 5 and Figure 6.

### B.5.3 Baseline Models for Metal Artifact Removal

**LI**   One straightforward way for reducing metal artifacts is to complete or inpaint the metal-affected missing regions in sinogram directly through linear interpolation (Kalender et al., 1987). This method does not need any network training. However, the imperfect completion of sinogram may introduce secondary artifacts to the reconstructed image. In our experiments setting, to fit for the practical applications in real world, we assume the ground truth metal trace and mask information are unknown, which can only be estimated by a rough thresholding in artifacts-affected images. We use the estimated metal mask and metal trace for linear interpolation baseline.

**cGANMAR**   Wang et al. (2018) proposed a conditional generative adversarial network (cGAN)-based approach for metal artifacts reduction (MAR) in CT. Specifically, cGANMAR network learns the mapping directly from the artifacts-affected CTs to artifacts-free CTs through refinement in image space. The cGANMAR model was implemented using PyTorch (Paszke et al., 2019) and was trained with the batchsize of 64 and the learning rate of 0.0001. The network was trained with 400 epochs.

**SNMAR**   Yu et al. (2020) proposed a sinogram completion neural network (SinoNet) to recover the metal-affected projections. Especially, it leveraged the learning in both sinogram domain and image domain by using a prior network to generate a good prior image to guide sinogram learning. Note that in original setting, SNMAR required linear interpolated sinogram and CT as inputs and used ground truth metal trace and mask information to generated them. But in our method, we assume the ground truth metal trace and mask information are unknown according to practical scenario and estimate it by a rough thresholding, which will introduce estimation errors. In SNMAR experiments, we still follow the original setting to guarantee the best performance of this baseline method for a strong comparison. We trained the SNMAR using the batchsize of 64 and the learning rate of 0.0001, with a total of 100 training epochs.

