# OpenReview forum: "Solving Inverse Problems in Medical Imaging with Score-Based Generative Models"
_ICLR.cc/2022/Conference — ICLR 2022 Poster_

### Official Review · Reviewer_5fGt · 2021-10-27

**Correctness:** 3
**Technical Novelty And Significance:** 2
**Empirical Novelty And Significance:** 3
**Recommendation:** 8
**Confidence:** 4

**Main Review:**

Strengths:

The overall idea - combine diffusion-based generative models with inverse problems - is well presented and the paper is easy to read. Compared to existing methods in this research direction, there is some novelty, mainly in the adapted sampling approaches.

The experimental results of the paper suggest that the proposed combination of unsupervised denoising score matching of a variance exploding SDE along with the novel sampling approach yields superior results and even outperforms supervised methods on some problems such as metal artifact removal.

Weaknesses:

The definition of inverse problems in section 2.1 is quite restrictive. The Dirac delta measurements distribution prevents any incorporation of real-world noise typically observed in inverse problems in medical imaging. For instance, in MRI frequently a Gaussian measurement noise is assumed, while the electron and photon noise in CT measurement can be described by a mixture of Gaussian and Poisson noise. Hence, real-world problems do not fit this restrictive definition of inverse problems.

All proofs (Proposition 1,  Lemma 1, and Theorem 1) are trivial and do not yield new theoretical insights.

As already mentioned above, the core novelty of the proposed paper is the additional sampling step performed in Eq. (5) and defined in Theorem 1. From an optimization point of view, the update rule in Theorem 1 implements a proximal mapping, which is very frequently used in MRI reconstruction [1]. This proximal mapping minimizes the quadratic energy
$$
\min_z \frac{1}{2}\Vert z-x\Vert_2^2 + \frac{\tau}{2}\Vert\text{diag}(\Lambda) T z - y\Vert_2^2
$$
for $\tau>0$ and reads as
$$
z = T^\top \left(\frac{Tx + \tau \text{diag}(\Lambda) y}{I + \tau \text{diag}(\Lambda)}\right),
$$
using the assumption $T^\top T=I$, which is fulfilled for a proper choice of $T$ in MRI and CT reconstruction problems. The equivalence of the proximal mapping and the mapping defined in Theorem 1 can be shown by using $I = (I-\Lambda)+\Lambda$ and the fact that $\frac{1}{1+\tau}+\frac{\tau}{1+\tau}$ is a convex combination. Clearly, the proximal map minimizes an energy that employs a quadratic penalization of the data fidelity, which implicitly leads to a Gaussian noise assumption of the measurement process. Consequently, the limited assumptions discussed in the first weakness are actually not relevant.

The general approach is motivated by an explicit discretization scheme of the backward SDE. However, proximal maps are related to implicit discretizations schemes. So, is the resulting algorithm a mixed discretization (semi-explicit)? What is the corresponding forward SDE? More details w.r.t. this regard would clearly improve the quality of the paper since the SDE perspective is an integral part of the entire work.

Since the choice of $\lambda$ often influences the reconstruction quality strongly, more details on determining this hyperparameter would be helpful.

In section 3.2 at the end of the second paragraph, the dimensions of $A\in\mathbb{R}^{m\times n}$ and $z\in\mathbb{R}^{m}$ do not match in the proposed sampling. Please, correct this.

The paper misses many implementation details. For instance, the learning problem (model, loss, optimization) is not explicitly defined.

[1] Hammernik, K., Schlemper, J., Qin, C., Duan, J., Summers, R. M., & Rueckert, D. (2021). Systematic evaluation of iterative deep neural networks for fast parallel MRI reconstruction with sensitivity‐weighted coil combination. Magnetic Resonance in Medicine.

**Summary Of The Paper:**

The manuscript applies denoising score matching to linear inverse problems to solve compressed sensing problems in medical imaging, such as angular-undersampled CT and accelerated MRI reconstruction. Throughout the paper, the observed measurements $y$ are considered noise-free, which is reflected by a Dirac measurement distribution. To train a score function, a variance exploding SDE is considered as in previous works, e.g. Song et al. (2021).
During inference, the authors incorporate a weighted projection onto measurement samples unlike previous approaches, which consider the gradient of the data distribution. This projection can be implemented in various sampling approaches such as annealed Langevin dynamics or predictor-corrector schemes.
The numerical results indicate that unsupervised score matching methods are well suited for inverse problems in imaging and yield superior generalization performance.

**Summary Of The Review:**

Overall I think the manuscript is marginally below the acceptance threshold due to unnecessary limitations of the approach and missing theoretical justification of the proposed discretization scheme of the reverse SDE despite the outstanding numerical results.

---

> ### Author Response · Authors · 2021-11-14
> **Thank you for the questions and feedback**
>
> Thank you for the detailed review and thoughtful feedback. We would like to stress that our approach is fully unsupervised—the models are all trained without knowing the measurement process, which is different from methods discussed in [2]. To the best of our knowledge, this is the first time that an unsupervised learning approach can outperform supervised counterparts on sparse-view CT and metal artifact removal in CT images.
>
> Below we address specific questions.
>
> **Q: Definition of the inverse problem in Section 2.1 is restrictive.**
>
> A: Thanks for the feedback. We have updated Section 2 and Section 3 to allow noisy measurements in our formulation. As you pointed out, our equation (8) is similar to a proximal optimization step. It can account for noise in measurements by using an appropriate $\lambda$. Our data for metal artifact removal have synthesized metal artifacts following the simulation procedure in [1], which explicitly adds Gaussian and Poisson noise. Our outstanding numerical results on this task thus demonstrate that our approach can indeed handle noise properly.
>
> **Q: Proofs of Proposition 1, Lemma 1, and Theorem 1 do not yield new theoretical insights.**
>
> A: We would like to emphasize that the goal of our paper is to provide an unsupervised technique with outstanding performance on multiple medical imaging tasks at the same time, including sparse-view CT, metal artifact removal for CT, and accelerated MRI. We agree with the reviewer that Proposition 1, Lemma 1, and Theorem 1 may not provide new theoretical insights for understanding inverse problems—we do not intend to. They are indispensable, however, for motivating and rigorously defining our method in Section 3.2 and Algorithm 2.
>
> **Q: Connection between Theorem 1 and proximal optimization.**
>
> A: Thank you for pointing out this connection. We agree that our optimization problem in equation (8) has the same intuition as performing one step of proximal optimization, and we have updated our paper to mention this. Please correct us if we are wrong, but we do not think your derivation for $z$ can be directly used in our setting because $T^\top T = I$ does not hold for CT applications, where $T$ corresponds to the Radon transform. In fact, one important motivation of our optimization problem in equation (8) is to sidestep the requirement $T^\top T = I$ for sparse-view CT and metal artifact removal in CT images.
>
> We do not think our algorithm can be interpreted as a mixed discretization scheme in a rigorous manner. According to our understanding, proximal optimization corresponds to implicit discretization of a gradient descent procedure for finding the MAP estimate of the posterior distribution $p(x \mid y)$. The SDE solver corresponds to an explicit discretization of a stochastic sampling process for $p(x)$. We do not see an immediate way to merge an optimization method and a sampling method into a unified scheme.
>
> In general, there are two important differences between our score-based method and existing iterative deep neural networks based on unrolling proximal optimization steps in [2]. First, our score-based approach aims to sample from $p(x \mid y)$, while methods in [2] aim to solve $\arg\max_x \log p(x \mid y)$. Second, our approach is fully unsupervised, and our models are trained without any knowledge of the measurement process (no data consistency layers). Data consistency is enforced only at the test time through our optimization step in equation (8). In contrast, methods in [2] require $y$ for training data consistency layers.
>
> **Q: More details on determining this hyperparameter $\lambda$ would be helpful.**
>
> A: We provided details in Appendix B.4. The hyperparameter $\lambda$ is determined by running 100 steps of Bayesian optimization on a validation dataset, and does not require manual tuning. We have updated Section 3 to mention this explicitly in the main text, and included a list of optimal $\lambda$ used in our experiments in Appendix B.4.
>
> **Q: Dimensions do not match in the proposed sampling.**
>
> A: Thanks for spotting the typo. We have corrected this in the revision.
>
> **Q: Missing implementation details.**
>
> A: Thanks for the feedback. We will release code and checkpoints for reproducibility upon paper acceptance. We have also updated Section 2 to include an explicit description on how to train our score-based models. As noted in Appendix B.4, other training details are exactly the same as [3].
>
> **References:**
>
> [1] Yu, L., et al. (2020). Deep sinogram completion with image prior for metal artifact reduction in CT images. IEEE Transactions on Medical Imaging, 40(1), 228-238.
>
> [2] Hammernik, K., et al. (2021). Systematic evaluation of iterative deep neural networks for fast parallel MRI reconstruction with sensitivity‐weighted coil combination. Magnetic Resonance in Medicine.
>
> [3] Song, Y., et al. (2020). Score-based generative modeling through stochastic differential equations. In ICLR 2021.

---

> > ### Comment · Reviewer_5fGt · 2021-11-18
> > **Feedback Revision**
> >
> > Thank you for your detailed reply and the accurate responses to my questions.
> >
> > The assumption $T^\top T$ could also be extended to the CT setting when the filtering step of (FBP) is evenly distributed to $T$ and $T^\top$. However, this requires a preprocessing of the observations $y$.
> >
> > In the revised paper in Section 3.2, you could consider citing a textbook on convex optimization [1,2], where proximal mappings are described in more detail.
> >
> > [1] Nesterov, Y. (2003). Introductory lectures on convex optimization: A basic course (Vol. 87). Springer Science & Business Media.
> >
> > [2] Boyd, S., Boyd, S. P., & Vandenberghe, L. (2004). Convex optimization. Cambridge university press.

---

> > > ### Author Response · Authors · 2021-11-19
> > > **Thanks for the feedback!**
> > >
> > > We have updated the manuscript to include the suggested citations. Thank you again for the feedback.

---

> > > ### Comment · Reviewer_5fGt · 2021-11-19
> > > **Proximal mappings**
> > >
> > > One point that might not have been clear in my review is that the assumption $T^\top T$ is not essential for proximal mappings. Hence, general inverse operators such $T^{-1}$  typically also occur in proximal mappings.

---

> > > > ### Author Response · Authors · 2021-11-19
> > > > **Thanks for the clarification**
> > > >
> > > > Thanks for the additional clarification. We agree with you that the assumption $T^\top T = I$ is not essential for defining a proximal mapping. Typically, a proximal mapping is defined as $prox_f(v) = \arg\min_{x} f(x) + \lambda || x- v||^2$, where $f$ is a convex function. The optimization problem in our equation (8) amounts to
> > > > $$
> > > > \arg\min_x d(x, S)^2 + \frac{1-\lambda}{\lambda} || x - v ||_T^2,
> > > > $$
> > > > where $d(x, S)$ denotes the distance between $x$ and the hyperplane $S := \\{ u | Au = y \\}$, measured in the $|| \cdot ||_T$ norm. Since $S$ is a convex set, $d(x, S)$ is a convex function of $x$, and therefore $d(x, S)^2$ is a convex function. As a result, our equation (8) satisfies the definition of a proximal mapping. Our revised manuscript has already adopted the proximal optimization interpretation for equation (8).
> > > >
> > > > Please kindly let us know whether this has addressed your remaining concerns of our paper. Happy to discuss more if you have additional concerns.

---

> ### Author Response · Authors · 2021-11-29
> **Follow up before the discussion is closed**
>
> Thank you again for your feedback. As we are approaching the end of the discussion period, we would like to ask whether our revision and response have addressed your initial concerns about our work. We are more than happy to clarify and discuss more should you have additional questions.

---

> > ### Comment · Reviewer_5fGt · 2021-11-30
> > **Thanks for the reply**
> >
> > Thank you for the detailed responses and the discussions. My concerns were addressed properly and the manuscript was edited accordingly. As a result, I believe that the manuscript provides a solid contribution to the field.

---

> > > ### Author Response · Authors · 2021-11-30
> > > **Thank you for the positive feedback**
> > >
> > > We are very glad that you liked our response and revision, and you think "the manuscript provides a solid contribution to the field". Given that we addressed your primary concerns raised in the review, we would kindly ask you to adjust your review score while taking the rebuttal into account. Thank you for the consideration.

---

### Official Review · Reviewer_fqS4 · 2021-11-01

**Correctness:** 4
**Technical Novelty And Significance:** 3
**Empirical Novelty And Significance:** 3
**Recommendation:** 6
**Confidence:** 3

**Main Review:**

The paper establish a new approach approach to solve linear inverse problems with score based generative models. In particular, given an unconditional score based generative model it is diffucult to sample directly from the conditional distribution $p(x_t|y)$ as it requires solving the conditional backward SDE. Instead, the authors propose to utlize the linear relationship between $x$ and $y$, and introduce a coupled stohcastic process $y_t$. Sampling can be done by adding an additional step that enforcing the linear relation between $x_t$ and $y_t$ at each sampling step.

The paper is well written and the idea is concise and novel. In current form it can only be applied to linear inverse problems, but I think there are certainly important applications of linear inverse problems so the contribution of this paper is substantial. I am not very familiar with the experimental evaluation of medical imaging, but I do see the paper making significant improments over previous methods based on paired training or generative models.

I have several concerns:

1. I think the biggest issue of the paper is lacking an motivation for the method. The method part of the paper can be re-organize a little bit to make the presentation clear. For example, the proposition 1 is somewhat absurd, as at the begining I do see why we need such a decomposition. The purpose of the extra iteration function is clear after reading it carefully, but I find it difficult to read at first look. I would suggest to first give a high level overview of the approach before section 2.1, that briefly introduces all the components of this approach and how to make them work, and them go through the details.

2. Since score-based model also has the probability flow ODE formulation, and it can be sampled by solving the ODE where the randonmess only come from initialization. Can the probability flow ODE formulation be used in this task? In addition, maybe you can also discuss previous approch that uses normalizing flow as prior for the solving inverse problems such as [1].

3. Let me know if I missed anything, but what is the sampling formuation for the item "Score SDE" in table 1? Do you train an explicit conditional score based model? In general I think more detailed introduction to each item in the table is needed, espeically for the item "Score SDE" which is highly related.

4. Also, do you include a study on if the result is sensitive to $\lambda$, the coefficient for the linear constraint?

[1] Invertible generative models for inverse problems: mitigating representation error and dataset bias. https://arxiv.org/abs/1905.11672

**Summary Of The Paper:**

The paper introduces a method to use score-based generative model as a powerful prior when solving linear inverse problems in medical imaging. Compared to previous work that applied score models to inverse problems, this paper proposes a new conditional sampling approach.

**Summary Of The Review:**

The paper is well written, with a clean idea and strong empirical results. There are some issues with the presenttaion. I will give a weak accept.

---

> ### Author Response · Authors · 2021-11-14
> **Thank you for the questions and feedback**
>
> Thank you for the detailed review and thoughtful feedback. Below we address specific questions.
>
> **Q: Lacking motivation for the method.**
>
> A: Thank you for the feedback. As suggested, we have rewritten Section 3 to include a high-level overview of the approach before Section 3.1. We additionally included a new schematic figure to illustrate our approach and provided more intuitions (see Figure 3).
>
> **Q: Can the probability flow ODE formulation be used in this task?**
>
> A: Yes, the probability flow ODE sampler also satisfies our definition of iterative sampling methods in equation (5) and can thus be used in our approach. We have updated Section 3 to mention this explicitly. Although the probability flow ODE is a continuous normalizing flow, our approach is quite different from inverse problem solving with normalizing flow priors in that (i) we aim to sample from $p(x \mid y)$, not finding $\arg\max_x \log p(x \mid y)$; and (ii) the probability flow ODE is not a normalizing flow anymore after discretization by ODE solvers and our additional sampling step in equation (6).
>
> **Q: Clarification of the “Score SDE” approach in Table 1.**
>
> A: We included more description of “Score SDE” in the “Unsupervised learning baselines” paragraph of Section 4. As an unsupervised learning technique, “Score SDE” does not train an explicit conditional score-based model. Instead, it leverages an unconditional score-based model to form a crude approximation to the conditional score function. The method which we call “Score SDE” has only been proposed as a theoretical possibility in Appendix I.4 of [1], and has never been evaluated empirically prior to our work.
>
> **Q: A study on whether the result is sensitive to $\lambda$.**
>
> A: As described in Appendix B.4, the value of $\lambda$ is automatically tuned by running 100 steps of Bayesian optimization on a validation dataset. Because there is no need to tune this hyperparameter by hand, we believe it is less of an issue for practical application of our approach. Empirically, we observed that the results are not sensitive to $\lambda$ for sparse-view CT and accelerated MRI if $\lambda$ is close to 1. For metal artifact removal, $\lambda$ needs to be smaller and Bayesian optimization finds that a value around $0.2$ is about optimal. We have provided a list of $\lambda$ values used in our experiments in Appendix B.4.
>
> **References:**
>
> [1] Song, Y., Sohl-Dickstein, J., Kingma, D. P., Kumar, A., Ermon, S., & Poole, B. (2020). Score-based generative modeling through stochastic differential equations. In International Conference on Learning Representations.

---

> ### Author Response · Authors · 2021-11-29
> **Follow up before the discussion is closed**
>
> Thank you again for your feedback. As we are approaching the end of the discussion period, we would like to ask whether our revision and response have addressed your initial concerns about our work. We are more than happy to clarify and discuss more should you have additional questions.

---

> > ### Comment · Reviewer_fqS4 · 2021-11-29
> > **Thanks for the reply**
> >
> > Thanks for the reply from the authors. My concerns have been addressed, and I would like to keep my score based on my judgement on the significance and novelty of this paper.

---

### Official Review · Reviewer_YBf4 · 2021-11-02

**Correctness:** 3
**Technical Novelty And Significance:** 3
**Empirical Novelty And Significance:** 3
**Recommendation:** 6
**Confidence:** 3

**Main Review:**

Strengths:
1) The paper targets an interesting, important, but challenging problem in medical image reconstruction, which could be attractive to ICLR participants.

2) Although I do not fully understand the proposed method (see the weaknesses below), it seems to be technically sound and efficient.

3) The experimental results demonstrate the effectiveness of the proposed method, which shows comparable or even better results compared to supervised methods.

Weaknesses:
1) The writing of the method section could be improved to make it easier to understand. What is the connection between Section 3.1 and 3.2? How to obtain T? Predfined transformations? How to compute P(\Lambda)? T and P(\Lambda) are essential to calculate A, while they are not clearly explained in the paper.

2) Maybe I missed the point, but what is the main difference between GAN-based methods and the score-based generative models? In the SDE model, is there a time-step size that should be predefined? If so, will it affect the generation result? How about the proposed method? How to set the step size? 1/N? how to choose N?

3) The paper claims the efficiency of the method, then what is the computational cost of the proposed method?

**Summary Of The Paper:**

This paper provides an unsupervised approach to solve the inverse problem for reconstructing medical CT and MRI scans using score-based generative models. The proposed method was evaluated on the LIDC and BraTS datasets. Compared to existing supervised and unsupervised approaches, the proposed method demonstrates comparable or better performances in terms of PSNR and SSIM.

**Summary Of The Review:**

Although the paper has some limitations, it presents good technical contributions. Therefore, I recommend the weak acceptance for this paper.

---

> ### Author Response · Authors · 2021-11-14
> **Thank you for the questions and feedback**
>
> Thank you for the detailed review and thoughtful feedback. Below we address specific questions.
>
> **Q: The writing of the method section could be improved.**
>
> A: Thank you for the feedback. We have rewritten Section 3 to highlight the connection between Section 3.1 and 3.2. In Section 3.1, we have a more detailed description on how to obtain $T$, and how to compute $\mathcal{P}(\Lambda)$. In particular, $T$ is the Radon transform and the spatial Fourier transform for CT and MRI applications respectively, which can be obtained efficiently with FFT-based algorithms. $\mathcal{P}(\Lambda)$ subsamples the outputs of $T$ according to a task-dependent mask (see Figure 2 for illustrations).
>
> **Q: What is the main difference between GAN-based methods and score-based generative models?**
>
> A: GAN-based methods [1] estimate the **mode** of $p(x \mid y)$, whereas score-based approaches produce approximate **samples** from $p(x \mid y)$. As a result, score-based methods can quantify the uncertainty of reconstructions by generating and comparing multiple samples, and have many other advantages as corroborated by [2] and [3]. To the best of our knowledge, GAN-based methods have never found success for unsupervised image reconstruction on clinical CT/MRI data. In contrast, we have achieved comparable or better performance than supervised techniques regarding sparse-view CT, metal artifact removal for CT, and accelerated MRI on multiple clinical datasets.
>
> **Q: Clarification of the SDE model.**
>
> A: We have more information in Appendix B.4. There are several hyperparameters in the SDE model that affect results and require tuning, including step size and the number of steps $N$. In our experiments, we automatically tune these parameters by running 100 steps of Bayesian optimization on a validation dataset. We set $N=1000$, and provide all other hyperparameters in Appendix B.4.
>
> **Q: Efficiency of the method.**
>
> A: Compared to other score-based approaches, our method does not require computing the SVD of the linear measurement operator $A$, which is prohibitively expensive for many tasks in medical imaging including CT reconstruction. The major bottleneck of computation in our approach is the evaluation of the score model. Generating one mini-batch of images requires evaluating the score model for 2000 times in our setting, which may take around 10 minutes on Tesla V100 GPUs.
>
> **References:**
>
> [1] Bora, A., Jalal, A., Price, E., & Dimakis, A. G. (2017, July). Compressed sensing using generative models. In International Conference on Machine Learning (pp. 537-546).
>
> [2] ​​Jalal, A., Karmalkar, S., Dimakis, A. G., & Price, E. (2021). Instance-Optimal Compressed Sensing via Posterior Sampling. In International Conference on Machine Learning.
>
> [3] Kawar, B., Vaksman, G., & Elad, M. (2021). SNIPS: Solving Noisy Inverse Problems Stochastically. In Neural Information Processing Systems.

---

> ### Author Response · Authors · 2021-11-29
> **Follow up before the discussion is closed**
>
> Thank you again for your feedback. As we are approaching the end of the discussion period, we would like to ask whether our revision and response have addressed your initial concerns about our work. We are more than happy to clarify and discuss more should you have additional questions.

---

> > ### Comment · Reviewer_YBf4 · 2021-11-30
> > **Thanks for the reply.**
> >
> > Thank you for the feedback, which clarifies my concerns. I'd like to keep my score unchanged.

---

### Author Response · Authors · 2021-11-14
**A summary of updates**

We would like to thank all reviewers for providing high quality reviews and constructive feedback that have improved the paper. We are encouraged that reviewers think our paper “targets an interesting, important, but challenging problem in medical image reconstruction” (Reviewer YBf4), is “technically sound and efficient” (Reviewer YBf4), “the contribution of this paper is substantial” (Reviewer fqS4); and “the paper is well written and the idea is concise and novel” (Reviewer fqS4).

We have updated our draft to further improve the writing and incorporate suggestions from reviewers, extended the appendix with more details for reproducibility, and will be releasing code and model checkpoints. Below, we summarize major changes made in the updated submission.

**A. Removing the noiseless assumption in the inverse problem formulation.**

In response to Reviewer 5fGt, we have updated Section 2 and 3 to allow noisy measurements in inverse problem solving.

**B. Rewriting the method section.**

We have rewritten Section 3 to address the concerns of Reviewer YBf4 and fqS4. Specifically, we emphasized the connection between Section 3.1 and 3.2, provided an explicit motivation for the decomposition given in Proposition 1, included a high-level overview of our approach before Section 3.1, and created a new schematic figure (Figure 3) to provide more intuitions for our approach.

**C. More implementation details.**

We have updated Section 2 to include a detailed description of the training process of our score models. We also expanded the appendix to include a list of optimal hyperparameters, and provided a more detailed explanation of how we used Bayesian optimization to automatically tune those hyperparameters. We will release the source code and model checkpoints upon acceptance.

---

### Decision · Program_Chairs · 2022-01-20

**Decision:**

Accept (Poster)

**Comment:**

This is an interesting paper on improving score-based conditional sampling and its use in solving inverse problems. The current method of sampling from NCSNv2 is somewhat inefficient and the authors propose a different SDE that seems to work better for conditional generation.

The paper is applied to Computational imaging and MRI and shows very good results and reasonable comparisons with the recent state of the art. One limitation is that the measurement process is artificial and ignores specifics of MRI (real measurements and multi-coils would strengthen the paper). In any case since this is a fundamental methods paper with a solid technical innovation on score-based sampling, I recommend acceptance.